# Training Overparametrized Neural Networks in Sublinear Time

## Abstract

The success of deep learning comes at a tremendous computational and energy cost, and the scalability of training massively overparametrized neural networks is becoming a real barrier to the progress of artificial intelligence (AI). Despite the popularity and low cost-per-iteration of traditional backpropagation via gradient decent, stochastic gradient descent (SGD) has prohibitive convergence rate in non-convex settings, both in theory and practice.

To mitigate this cost, recent works have proposed to employ alternative (Newton-type) training methods with much faster convergence rate, albeit with higher cost-per-iteration. For a typical neural network with $m = \text{poly}(n)$ parameters and input batch of $n$ datapoints in $\mathbb{R}^d$, the previous work of Brand et al. (2021) requires $\sim mnd + n^3$ time per iteration. In this paper, we present a novel training method that requires only $m^{1-\alpha}nd + n^3$ amortized time in the same overparametrized regime, where $\alpha \in (0.01, 1)$ is some fixed constant. This method relies on a new and alternative view of neural networks, as a set of binary search trees, where each iteration corresponds to modifying a small subset of the nodes in the tree. We believe this view would have further applications in the design and analysis of deep neural networks (DNNs). We conclude a discussion of lower bound for the dynamic sensitive weight searching data structure we make use of, showing that under SETH or OVC from computational complexity, one cannot substantially improve our algorithm.

## 1 Introduction

Deep learning technology achieves unprecedented accuracy across many domains of AI and human-related tasks, from computer vision, natural language processing, and robotics. This success, however, is approaching its limit and is largely compromised by the computational complexity of these resource-hungry models. State-of-art neural networks keep growing larger in size, requiring giant matrix operations to train billions of parameters Devlin et al. (2018); Radford et al. (2019); Brown et al. (2020); Chowdhery et al. (2022); Zhang et al. (2022); ChatGPT (2022); OpenAI (2023). This barrier is exacerbated by the empirical phenomenon that *overparametrization* in DNNs Jacot et al. (2018) keeps improving model accuracy, despite the danger of overfitting Nakkiran et al. (2021), motivating the design of complex networks which need to train billions of parameters. As such, scalable training of deep neural networks is a major challenges of modern AI Wu et al. (2022); Spring & Shrivastava (2017).

Training a neural network can be broadly viewed a greedy iterative process, starting from an initial set of weight matrices (one per layer of the network). In each iteration, the algorithm chooses a (possibly complicated) rule for updating the value of current weights $W_i$ based on the training data, yielding the new weight matrices $W_{i+1}$. The total running time of DNN training is generally composed of two parts: The *number of iterations* (i.e., convergence rate) and the *cost-per-iteration* (i.e., CPI). A long line of research in convex and non-convex optimization has focused on the former question Khachiyan (1980); Karmarkar (1984); Vaidya (1987); Renegar (1988); Vaidya (1989); Madry (2013); Lee & Sidford (2014); Madry (2016); Lee et al. (2019); Jiang et al. (2020); Huang et al. (2022); Shi et al. (2022); Deng et al. (2023; 2024); Shi et al. (2024); Gu et al. (2024). This paper's focus is on the latter question.

The most popular iterative method for training DNNs is via *stochastic gradient descent* and its regularized variations Li & Liang (2018); Du et al. (2019); Allen-Zhu et al. (2019b;c); Song & Yang (2019); Wu et al. (2019); Deng et al. (2024). The popularity of this method is justified, to a great extent, by the simplicity and fast CPI. Calculating the gradient of the loss function is linear in the dimension of the gradient in each iteration, especially with mini-batch sampling Hardt et al. (2016); Cai et al. (2019)). Alas, the theoretical convergence rate (number of iterations) of first-order methods is dauntingly slow in *non-convex* landscapes due to pathological curvatures ($\Omega(\text{poly}(n)\log(1/\epsilon))$ for reducing the training error below $\epsilon$ in overparametrized networks, see e.g., Zhang et al. (2019)).

A recent line of work proposed to mitigate this drawback by replacing (S)GD with *second-order* (Newton-type) methods, which exploit information of the Hessian (curvature) of the loss function, and are proven to converge dramatically faster, at a rate of $O(\log(1/\epsilon))$ iterations, which is *independent* of the input size Martens & Grosse (2015); Zhang et al. (2019). In contrast, Newton methods have a high CPI. since they need to compute the *inverse* of Hessian matrix, which is dense and changes dynamically. The recent works of Cai et al. (2019); Zhang et al. (2019) showed that this computational bottleneck can be mitigated for overparametrized DNNs ($m = \text{poly}(n)$) with smooth (resp. ReLU) activations, and presented a Gauss-Newton (resp. NGD) training algorithm with $O(mn^2)$ training time per iteration. Here $m$ is the number of neurons. We let $n$ be the number of inputs. This runtime was further improved in the work of Brand et al. (2021), who showed how to implement the Gauss-Newton algorithm in $O(mnd + n^3)$ time per iteration, which is *linear time* in the network size, assuming $m \gtrsim n^2$ (as the dimensions of the Jacobian matrix of the loss is $\Theta(mnd)$ without simplifying assumptions Martens & Grosse (2015)).

## 1.1 OUR RESULT – AN UPPER BOUND

It is tempting to believe that linear-time per iteration Brand et al. (2021) is unavoidable – For a network with $m$ neurons and a training set of $n$ points in $\mathbb{R}^d$, each iteration spends at least $\sim nmd$ time to go through each training datapoint and each neuron. Indeed, this was a common feature of all aforementioned training methods.

Nevertheless, in this paper we present a novel training method with *sublinear* cost per iteration in the network size, while retaining *the same convergence rate* (number of iterations) as the prior state-of-art methods Brand et al. (2021); Zhang et al. (2019); Cai et al. (2019). More formally, let $f : \mathbb{R}^d \to \mathbb{R}$ be a neural network

$$f(x) := \sum_{r=1}^{m} a_r \cdot \phi(\langle w_r, x \rangle - b)$$

with bias $b > 0$, $a \in \{\pm 1\}^m$, each $w_r \in \mathbb{R}^d$, for all $r \in [m]$. Our main resuilt is as follows.

**Theorem 1.1** (Main Result, Informal)**.** *Suppose there are $n$ training data points in $\mathbb{R}^d$. Let $f_{m,n}$ be a sufficiently wide two-layer* ReLU *NN with $m = \text{poly}(n)$ neurons. Let $\alpha \in (0.01, 1)$ be some fixed constant. Let $\epsilon \in (0, 0.1)$ be an accuracy parameter. Let $\mathcal{T}(\epsilon)$ denote the overall time for shrinking loss down to $\epsilon$. There is a (randomized) algorithm (Algorithm 1) that, with probability $1 - 1/\text{poly}(n)$, reduces the training error by $1/2$ in each iteration (note that $f_t$ is $f_{m,n}$ at time $t$)*

$$\ell_2-\text{loss}(f_{t+1}, y) \leq \frac{1}{2} \cdot \ell_2-\text{loss}(f_t, y)$$

*in amortized cost-per-iteration (*CPI*)*

$$\widetilde{O}(m^{1-\alpha}nd + n^3).$$

*The overall running time (including initialization) $\mathcal{T}(\epsilon)$ is*

$$O(mnd) + \widetilde{O}(m^{1-\alpha}nd + n^3)) \cdot \log(1/\epsilon).$$

*If the algorithm is allowed to use fast matrix multiplication (*FMM*), then the* CPI *becomes*

$$\widetilde{O}(m^{1-\alpha}nd + n^\omega),$$

*and the $\mathcal{T}(\epsilon)$ becomes*

$$O(mnd) + \widetilde{O}(m^{1-\alpha}nd + n^\omega) \cdot \log(1/\epsilon),$$

*where $\omega$ is the exponent of matrix multiplication, which is currently approximately equal to* 2.373.

*The randomness is from two parts: the first part is random initialization weights, and the second part is due to internal randomness of our algorithm.*

**Remark 1.2.** *Notice that the linear cost term $O(mnd)$ for merely computing the network's loss matrix, is only incurred once at the* initialization *of our training algorithm, whereas in Brand et al. (2021) and all prior work Cai et al. (2019); Zhang et al. (2019), this linear term is payed every iteration (i.e., $T \cdot mnd$ as opposed to our $T + mnd$). Our theorem therefore provides a direct improvement over Brand et al. (2021) when $m = \text{poly}(n)$.*

**Key Insight: DNNs as Binary-Search Trees**  Our algorithm is based on an alternative view of DNNs, as a *set of binary search trees*, where the relationship between the network's weights and a training data point is encoded using a binary tree: Each leaf represents the inner product of a neuron and the training data, and each intermediate (non-leaf) node represents the *larger* out of the left and right child. This simple yet new representation of neural networks turns out to enable fast training – The centerpiece of our result is an analysis proving that in each iteration, only a small subset $K$ of paths in this tree collection needs to be updated (amortized worst-case), due to the *sparsity* of activations. Consequently, we only need to update $nK \log m$ tree nodes per iteration. In the Technical Overview Section 4, we elaborate more on its details.

## 1.2 Our result – a lower bound

When it comes to efficiently maintaining and updating the weights, we design a special data structure supports the dynamic sensitive weight searching. The task is defined as follows.

**Definition 1.3** (Dynamic Sensitive Weight Searching (DSWS))**.** *We ask to design a data structure which supports the following procedures:*

- INIT($\{w_1, w_2, \cdots, w_m\} \subset \mathbb{R}^d, \{x_1, x_2, \cdots, x_n\} \subset \mathbb{R}^d$. *Given a series of weights $w_1, w_2, \cdots, w_m$ and datas $x_1, x_2, \cdots, x_n$, it preprocesses them.*

- UPDATE($z \in \mathbb{R}^d, j \in [m]$). *Given a new weight vector $z \in \mathbb{R}^d$ and index $j \in [m]$, it updates weight $w_j$ with $z$.*

- QUERY($i \in [n], \tau \in \mathbb{R}$). *Given a query index $i \in [n]$ and a threshold $\tau \in \mathbb{R}$, it finds all index $j \in [m]$ such that $\langle w_j, x_i \rangle \geq \tau$.*

We propose a data structure to solve DSWS with $\widetilde{O}(nd)$ time update and $\widetilde{O}(K_q)$ where $K_q := |\{j \in [m] \mid \langle w_j, x_i \rangle \geq \tau\}|$. The full detail can be found in Theorem B.1. By the sparsity guarantee, we have $|K_q| \leq m^{0.76}$, which leads to a query time of $\widetilde{O}(m^{0.76})$ and total time of $\widetilde{O}(m^{0.76}n)$ to query for all $i \in [n]$. In order to evaluate how far is our algorithm away from optimal, we provide a lower bound result for it.

**Theorem 1.4** (Lower Bound for DSWS, informal version of Theorem 7.3)**.** *Let $d = 2^{O(\log^* n)}$ and $m = \text{poly}(n)$. Then for every $\epsilon > 0$, assuming SETH or OVC, DSWS cannot achieve $O(n^{1-\epsilon})$ time of update and $O(m^{0.76}n^{1.24-\epsilon})$ time to query for all $i \in [n]$.*

This result shows that it is almost impossible to truly improve our algorithm. We provide the full discussion of this hardness in Section 7.

**Roadmap.**  We describe the organization of this work in next a few sentences. We state some related work in Section 2. We propose our main problem and present the tools we need to use in Section 3. In Section 4, we specifically overview the techniques used in this paper. In Section 5, we analyze the correctness of our algorithm, specifically, we prove the training loss converges. In Section 6, we analyze the running time of our algorithm. We provide the lower bound analysis in Section 7. In Section 8, we state our conclusion.

## 2 Related Work

**Speedup with high-dimensional search data structure.**  Advancements in high-dimensional search data structures allow for rapid identification of points within complex geometric query

regions (such as half-spaces and simplices). Presently, two primary methodologies are utilized in the construction of these structures. The first relies on Locality Sensitive Hashing (LSH) Indyk & Motwani (1998), designed to discover points nearby in terms of small $\ell_2$ distance Datar et al. (2004); Andoni & Razenshteyn (2015); Andoni et al. (2015; 2017); Razenshteyn (2017); Andoni et al. (2018); Backurs et al. (2019); Dong et al. (2020) or large inner product Shrivastava & Li (2014a;b; 2015) relative to a query point $q \in \mathbb{R}^d$ among a set of points $S \subseteq \mathbb{R}^d$. While LSH-based algorithms are fast in practice, they primarily support only approximate nearest neighbor queries. The alternative approach involves space partitioning data structures, such as partition trees Matoušek (1991); Matousek (1992); Agarwal et al. (1992); Afshani & Chan (2009); Chan (2010), k-d trees, range trees Chan & Tsakalidis (2017); Toth et al. (2017); Chan (2019), and Voronoi diagrams Agarwal et al. (1994); Chan (2000), which allow for exact location of points within the queried area.

**Over-parameterized Neural Networks.** Convergence through over-parametrization, where trainable parameters (m) significantly outnumber training data points (n, i.e., $m \gg n$), is a core aspect of deep learning. This setup helps to explain the adaptability of deep neural networks across diverse applications. Recent studies have focused on theoretically understanding the mechanisms behind deep learning convergence and generalization in this context Li & Liang (2018); Du et al. (2019); Allen-Zhu et al. (2019b;c); Arora et al. (2019a;b); Song & Yang (2019); Cai et al. (2019); Zhang et al. (2019); Cao & Gu (2019); Zou & Gu (2019); Oymak & Soltanolkotabi (2020); Ji & Telgarsky (2019); Lee et al. (2020); Huang et al. (2021); Zhang et al. (2020); Brand et al. (2021); Song et al. (2021b); Zhang (2022); Shi et al. (2022; 2024); Gu et al. (2024); Alman et al. (2024b). It is noted that as network width (m) increases, the behavior of neural networks aligns with a neural tangent kernel (NTK). Research shows that (stochastic) gradient descent ((S)GD) can effectively train wide networks starting from random initializations to achieve minimal training error in polynomial steps Jacot et al. (2018).

**Fine-grained Complexity and Orthogonal Vector Conjecture.** The Orthogonal Vector problem (OV) is a key issue in fine-grained complexity, posing the question: given sets $X, Y \subseteq \{0, 1\}^d$ of equal size n, are there vectors $x \in X$ and $y \in Y$ such that their dot product $\langle x, y \rangle = 0$? The advanced algorithm for this Abboud et al. (2014a); Chan & Williams (2016) operates in a time complexity of $n^{2-1/O(\log c)}$ for dimension $d = c \log n$, with $c \geq 1$, and as $d$ grows, its time complexity nears the trivial $n^2$. The orthogonal vector conjecture (OVC) posits a lower bound for OV when $d = \omega(\log n)$. Additionally, the Strong Exponential Time Hypothesis (SETH) suggests that the difficulty of k-SAT implies OVC. This conjecture is foundational for deriving conditional lower bounds for a range of significant problems that otherwise have polynomial-time solutions across several fields, including pattern matching Abboud et al. (2014b); Bringmann (2014a;b); Backurs & Indyk (2016); Bringmann & Mulzer (2016); Bringmann et al. (2017); Bringman & Künnemann (2018); Chen & Williams (2019), graph theory Roditty & Vassilevska Williams (2013); Abboud et al. (2018); Gao et al. (2018); Krauthgamer & Trabelsi (2018); Dalirrooyfard et al. (2022); Chan et al. (2022), and computational geometry Buchin et al. (2016); Rubinstein (2018); Williams (2018a); Chen (2018); Karthik & Manurangsi (2020). For further details, see the survey Williams (2018b).

## 3 PRELIMINARIES

### 3.1 MODEL FORMALIZATION

In this section, we formalize the NN model and the main problem of this paper. When there is no ambiguity, we will always use the notations in this section throughout the whole paper.

We first define the 2-layer ReLU activated neural network and its loss function.

**Definition 3.1** (2-layer ReLU activated neural network). *Suppose the dimension of input is* $d$, *the number of intermediate nodes (or hidden neurons) is* $m$, *the dimension of output is* $1$, *the batch size is* $n$ *and the shifted parameter is* $b$ ($b \geq 0$). *Then the weight of the first layer can be characterized by* $m$ $d$-*dimensional vectors* $w_1, w_2, \cdots, w_m$, *and the weight of the second layer can be characterized by* $m$ *scalars* $a_1, a_2, \cdots, a_m$. *For convenience, define* $W = [\, w_1^\top \; w_2^\top \; \cdots \; w_m^\top \,]^\top$ *and* $a = [\, a_1 \; a_2 \; \cdots \; a_m \,]^\top$, *given an input* $x \in \mathbb{R}^d$, *the 2-layer* ReLU *activated neural network outputs* $f(W, x, a) = \frac{1}{\sqrt{m}} \sum_{r=1}^m a_r \phi(\langle w_r, x \rangle)$ *where* $\phi(x) = \max\{x, b\}$ *is called shifted* ReLU *activation function.*

*For simplicity, we suppose the data is normalized, that is, $\|x\|_2 = 1$. This is natural in both practical machine learning, and machine learning theory.*

*We also suppose $a \in \{-1, +1\}^m$ is fixed throughout training. This is also natural in the area of theoretical deep learning Li & Liang (2018); Du et al. (2019); Allen-Zhu et al. (2019b;a); Song & Yang (2019); Brand et al. (2021); Zhang (2022).*

For more detailed formalizations, we refer to Section A.2 in the appendix.

## 3.2 PROBLEM DEFINITION

We formalize our main problem as follows.

**Definition 3.2** (Main problem). *The goal of this paper is to propose a training algorithm such that for an arbitrary 2-layer ReLU activated neural network defined in Definition 3.1, it converges with high probability, and the running time of each iteration is sublinear in $nmd$ (i.e. $o(nmd)$).*

## 4 TECHNICAL OVERVIEW

Here, we describe the outline of the main ideas required to prove Theorem 1.1.

**Key Ideas** Our algorithm relies on two simple but powerful observations about training 2-layer neural networks: The first observation is that the Jacobian matrix of the loss function is *sparse* – When weights are initialized randomly (with appropriately chosen bias parameter $b$), the fraction of nonzero entries in the Jacobi matrix is small. Let $c$ be some fixed constant in $[0.1, 1]$. We show that there is a choice of the parameter $b$ ensuring simultaneously that[1]

- For every input $x_i$, there are only $O(m^{1-c})$ activated neurons;
- The loss of each iteration is still at most a half of the loss of the last iteration.

Our second observation is that the *positions* of the nonzero entries in Jacobian matrix do not change much. This can be seen using the "gradient flow" equation (via Gauss-Newton method) $W_{t+1} = W_t - J_t^\top g_t$, where $g_t := \arg\min_g \|J_t J_t^\top g_t - (f_t - y)\|_2$. Since the Jacobian matrix is sparse, it is not hard to see that only a little fraction of the weights need to be modified, i.e., the change from $W_t$ to $W_{t+1}$ involves updating only a small number of entries.

These two observations suggests a natural "binary-search" type algorithm for updating the weight matrix in *sublinear time $o(nmd)$* per iteration.

**Threshold search data structure** We design a dynamic data structure for detecting and maintaining the non-zero entries of the Jacobian matrix $J$ of the network loss, as it evolves over iterations. Notice that whether an entry of $J$ is nonzero is equivalent to whether the inner product of an input $x_i$ and a weight $w_j$ is larger than $b$ (hence $\phi(w_j^\top x_i) > 0$).

Accordingly, for every input $x_i$ in a batch, our algorithm maintains a binary search tree $\mathcal{T}_i$ where each leaf stores the inner product of $x_i$ and a weight $w_j$, and every non-leaf node stores the the *maximum* of the values of its two children. In this way, non-zero entries can be found by searching, in all the trees $\{\mathcal{T}_i\}_{i \in [n]}$, from root to leaf and ignoring the unnecessary branches.

To implement this process efficiently, our data structure needs to support the following three operations (See Section B for the formal details): (1) **Initialization.** Given input vectors $x_1, \cdots, x_n$ and weight vectors $w_1, w_2, \cdots, w_m$ as input, it constructs $n$ binary trees $\mathcal{T}_1, \ldots, \mathcal{T}_n$ as described above, in $O(mnd)$ time. (2) **Updating of weights.** Taken an index $j \in [m]$ and a target value $z$, it replaces $w_j$ by $z$ in $O(nd + n \log m)$ time, as if initializing it with $w_1, w_2, \cdots, w_{j-1}, z, w_{j+1}, \cdots, w_m$ from scratch. (3) **Threshold Search Query.** Given an index $i$ and a threshold $\tau$ as input, our data structure rapidly finds all the weights $w_j$ which satisfies $\langle x_i, w_j \rangle \geq \tau$ in $O(K_q \log m)$ time, where $K_q$ is the number of satisfied weights. They can be used to find the nonzero entries of the Jacobian matrix $J$.

---

[1]We refer the readers to Section F for more details.

**A Fast DNN Training Algorithm** Using the above dynamic data structure, we design a fast neural network training algorithm (see Algorithm 1) composed of initialization and the (dynamic) training process. At initialization, it initializes the weight vector $W_0$ randomly.

The training process consists of maintaining *sparse-recovery* sketches Ailon & Chazelle (2006); Lu et al. (2013); Nakos & Song (2019), online regression, and implicit weight maintenance. The goal of the first two techniques is to efficiently solve the $t$-th iteration regression problem (cf. Brand et al. (2021)) $g_t := \arg\min_g \|J_t J_t^\top g - (f_t - y)\|,$. The idea of implicit-weight-maintenance (via our data structure) is to update weights using the information propagated by the loss function.

The details of these three tools can be summarized as follows:

- **Sketch maintenance** The goal of sketch computing is to eliminate the disastrous influence of the high dimension of $J_t^\top$ (it has $md$ rows) when solving regression problem in Eq. (4). Roughly speaking, in sketch computing, we find a sketch matrix $S$ with far smaller rows than $J_t^\top$ such that for any $d$-dimensional vector $x$, $\|SJ_t^\top x\|_2$ is very close to $\|J_t^\top x\|_2$. We show that sketch computing runs in $o(mnd)$ time.

- **Iterative regression solver** To speed-up the solution of the online regression problem (4), we show how to implement the iterative Conjugate-Gradient solver (a-la Brand et al. (2021)) *in sub-linear time* to find an approximate solution $g_t$ in time $o(mnd) + \widetilde{O}(n^3)$. We then prove that the (accumulated) approximation errors do not harm the convergence rate and precision in our analysis.

- **Implicit weight maintenance** The goal of implicit weight maintenance is to update weights according to the outcome of the iterative regression solver. Updating a single weight can be done by calling UPDATE once. With the result of iterative regression and the fact that only $m^{-c}$ (where $c$ is some fixed constant $c \in [0.1, 1]$) fraction of entries of $J_t$ are nonzero, we show that our algorithm finishes the update of weights in $o(mnd)$ time.

The details can be found in the pseudocode of Algorithm 5.

## 5 CONVERGENCE ANALYSIS

We focus on the convergence of our training algorithm in this section and leave the proof of running time in Section 6. Specifically, the goal of this section is to prove the following result, which implies that for the neural network randomly initialized at the beginning of our algorithm, the loss function converges linearly with high probability. This section only contains a proof sketch. For more detailed correctness analysis, we refer the readers to section D. Our main convergence result is the following:

**Theorem 5.1** (Formal version of Theorem 1.1, the convergence part)**.** *Let $m$ be the width of the NN. If $m = \Omega(\max\{\lambda^{-4}n^4, \lambda^{-2}n^2d\log(16n/\rho)\})$, then there is a constant $c' > 0$ so that our algorithm obtains $\|f_{t+1} - y\|_2 \leq 0.5 \cdot \|f_t - y\|_2$. It holds with probability $1 - \frac{5}{2}\rho - n^2 \cdot \exp(-m \cdot \min\{c'e^{-b^2/2}, \frac{R}{10\sqrt{m}}\})$ The randomness comes from two parts: the initialization of neural network and iterative algorithm itself.*

**Bounding the Function Value and Jacobian at the Initialization** We provide a lemma which shows that, with random initialization, as long as the 2-layer NN is wide enough, the norm of weight matrix, the initial predicted value and the Frobenius norm of the initial Jacobi matrix are all not large with high probability. We defer its proof into Section D.

**Lemma 5.2** (Informal version of Lemma D.1)**.** *Consider shifted* ReLU. *Suppose $m$ is the width of neural network. If $m = \Omega(d\log(16n/\rho))$, then we have the followingh olds with probability $1 - \rho/2$,*

- $\|W_0\|_2 = O(\sqrt{m})$.

- $\max_{i \in [n]} |f(W, x_i)| = O(1)$.

- $\max_{i \in [n]} \|J_{W_0, x_i}\|_F = O(1)$.

$G$ **does not move much when** $W$ **does not move much** We provide a lemma which proves that, as long as the 2-layer NN is wide enough, then with high probability that, for randomly initialized

weights $W_0$, if $W_0$ changes to $W$ after a small change, then the Gram matrix $G_W$ will not move much and the minimal eigenvalue of $G_W$ will also not move much. And We leave its proof in Section D.

**Lemma 5.3** (Shifted Perturbation Lemma, informal version of Lemma D.2). *Consider shifted ReLU with b. Let $b \geq 0$. Let $R_0 > 0$. Suppose $m \geq \Omega(1) \cdot \max\{b^2 R_0^2, n^2 R_0^2 \lambda^{-2}, n\lambda^{-1} \log(n/\rho)\}$, then with prob. $\geq 1 - \rho - n^2 \cdot \exp\left(- m \cdot \min\{c' e^{-b^2/2}, \frac{R_0}{10\sqrt{m}}\}\right)$, for any weight $W \in \mathbb{R}^{d \times m}$ satisfying $\max_{r \in [m]} \|w_r - w_r(0)\|_2 \leq R_0/\sqrt{m}$, the following holds: $\|G_W - G_{W_0}\|_F \leq \lambda/2$, and $\lambda_{\min}(G_W) \geq \lambda/2$. Note that $w_r$ is representing the $r$-th column of $W$.*

**Perturbed weights difference under shifted NTK**    We give a lemma which proves that, as long as the 2-layer NN is wide enough, then with high probability that, for randomly initialized weights $W_0$, if $W_0$ changes to $W$ after a small change, the each row $J_{W,x_i}$ of Jacobi matrix $J_W$ will not change much, and the Frobenius norm of $J_W$ will also not change much. We leave its proof in Section D.

**Lemma 5.4** (Informal version of Lemma D.3). *Suppose $R_0 \geq 1$ and $m = \widetilde{\Omega}(n^2 R_0^2)$. With probability at least $1 - \rho$ over the random initialization of $W_0$, the following holds for any set of weights $w_1, \ldots w_m \in \mathbb{R}^d$ satisfying $\max_{r \in [m]} \|w_r - w_r(0)\|_2 \leq R_0/\sqrt{m}$,*

- $\|W - W_0\| = O(R_0)$,

- $\|J_{W,x_i} - J_{W_0,x_i}\|_2 = \widetilde{O}(R_0^{1/2}/m^{1/4})$ *and* $\|J_W - J_{W_0}\|_F = \widetilde{O}(n^{1/2} R_0^{1/2}/m^{1/4})$,

- $\|J_W\|_F = O(\sqrt{n})$.

**Induction Hypothesis**    Finally, we're ready to prove our major theorem, Theorem 5.1. Note that we only need to prove the induction hypothesis described in definition 5.5, then Theorem 5.1 holds by mathematical induction. We divide the proof of this hypothesis into 2 parts and prove them in section E.1 and section E.2 respectively.

**Definition 5.5** (Induction hypothesis). *Define $R_0 \approx n/\lambda$. For any fixed $t$, if $\|f_t - y\|_2 \leq \frac{1}{2}\|f_{t-1} - y\|_2$ and $\max_{r \in [m]} \|w_r(t) - w_r(0)\|_2 \leq R_0/\sqrt{m}$. Then we have $\|f_{t+1} - y\|_2 \leq \frac{1}{2}\|f_t - y\|_2$ and $\max_{r \in [m]} \|w_r(t+1) - w_r(0)\|_2 \leq R_0/\sqrt{m}$.*

Formally, we describe the process of proving this hypothesis by the following Lemma 5.6, and specific proof can be seen in Section E.

**Lemma 5.6.** *Suppose initial weights $W_0$ satisfies the restriction of Lemma 5.2, 5.3 and 5.4, then the induction hypothesis described in Definition 5.5 holds.*

---

**Algorithm 1** Our training algorithm, informal version of Algorithm 5

---

1: **procedure** OURALGORITHM($X, \epsilon$)
2:     Initialization Step: randomly pick $W(0)$, $T \leftarrow \log(1/\epsilon)$, create a data structure
3:     Iterative Step: start with $t = 1$
4:         Step 1: Do the sketch computing, it forms matrix $S \in \mathbb{R}^{N \times n}$
5:         Implicitly write down the Jacobian matrix $J_t \in \mathbb{R}^{n \times md}$
6:         Choose sketch related parameters as Definition 6.2
7:         Find sketching matrix $S \in \mathbb{R}^{s_{\text{sketch}} \times md}$ of $J_t^\top$
8:         Step 2 Run an iterative regression algorithm with small size problem (size reduced by sketch)
9:         Find approximated solution $g_t$ of regression problem $\arg\min_g \|(J_t S^\top)(S J_t^\top)g - (f_t - y)\|$
10:        Step 3: Maintain the weight implicitly
11:        Update the weights $W_t$ to $W_{t+1}$
12:        Update the TS data structure using $W_{t+1}$
13:        Increment $t$ by 1
14: **end procedure**

---

## 6 RUNNING TIME ANALYSIS

This section focuses on analyzing the running time of our algorithm. It will show that when $m$ is large enough, the CPI is $o(nmd)$. We first present Theorem 6.1, our main running time result of the paper. For more proof details of the running time, we refer the readers to Section F. For simplicity of presentation, we use $o(m)$ and $o(mnd)$ in this section. In Section F, we explicitly compute time by $m^{1-\alpha}$ and $m^{1-\alpha}nd$ where $\alpha \in [0.01, 1)$ is some fixed constant. Our main running time result is the following:

**Theorem 6.1** (The running time part of Theorem 1.1). *The cost per iteration (CPI) of our algorithm is $\widetilde{O}(n^2 m^{0.76}d + n^3)$ or $\widetilde{O}(m^{1-\alpha}nd + n^3)$ by assuming $m$ is as large as $n^c$ without using FMM. The CPI of our algorithm is $\widetilde{O}(n^2 m^{0.76}d + n^\omega)$ or $\widetilde{O}(m^{1-\alpha}nd + n^\omega)$ by assuming $m$ is as large as $n^c$ with using FMM. Here $\alpha \in [0.1, 0.24], c \geq 8$ are two constant factors.*

*Proof.* Combining Lemma F.2, Lemma F.3 and Lemma F.4, the computation time of each iteration is

$$\widetilde{O}(n^2 m^{0.76}d) + \widetilde{O}(nm^{0.76}d + n^3) + O(n^2 m^{0.76}(d + \log m)) = \widetilde{O}(n^2 m^{0.76}d + n^3).$$

And if using FMM, similarly the running time is $\widetilde{O}(n^2 m^{0.76}d + n^\omega)$. By Theorem 5.1, we have: The time to reduce the training loss to $\epsilon$ is $\widetilde{O}((n^2 m^{0.76}d + n^3)\log(1/\epsilon))$. Taking advantage of FMM, the time is $\widetilde{O}((n^2 m^{0.76}d + n^\omega)\log(1/\epsilon))$. Further, for example, if $m = n^c$ where $c$ is some large constant, then $n^2 m^{0.76}d \leq nm^{1-\alpha}d$ where $\alpha \in [0.1, 0.24]$. Hence the time of each iteration is $\widetilde{O}(m^{1-\alpha}nd + n^3)$, and the time to reduce the training loss to $\epsilon$ is $\widetilde{O}((m^{1-\alpha}nd + n^3)\log(1/\epsilon))$. Taking advantage of FMM, the time is $\widetilde{O}((m^{1-\alpha}nd + n^\omega)\log(1/\epsilon))$. Thus we complete the proof. □

**Sketch Computing.** We provide the choice of sketching parameters in the following definition and give a lemma that analyzes the running time of the sketch computing process in Algorithm 5 under these parameters. It will imply that sketch computing is sublinear in $m$.

**Definition 6.2** (sketch parameters). *We choose sketch parameters in the following ways: $\epsilon_{\mathrm{sketch}} = 0.1, \delta_{\mathrm{sketch}} = 1/\operatorname{poly}(n), s_{\mathrm{sketch}} = n \operatorname{poly}(\epsilon_{\mathrm{sketch}}^{-1}, \log(n/\delta_{\mathrm{sketch}}))$*

**Lemma 6.3** (Step 1, sketch computing. Informal version of Lemma F.2). *The sketch computing process of Algorithm 1 (its formal version is Algorithm 5) runs in time $o(mnd)$.*

**Iterative regression.** We present a lemma that analyzes the running time of the iterative regression process in Algorithm 5. It implies that the running time of the iterative regression is sublinear in $m$.

**Lemma 6.4** (Step 2, running time of iterative regression. Informal version of Lemma F.3). *The iterative regression of Algorithm 1 (its formal version is Algorithm 5) runs in time $O(o(mnd)\log(n/\delta) + n^3)$. Taking advantage of FMM, it takes time $O(o(mnd)\log(n/\delta) + n^\omega)$, where $\omega$ is the exponent of matrix multiplication. Currently $\omega \approx 2.373$ Williams (2012).*

**Implicit weight maintenance.** We give a lemma that analyzes the running time of the implicit weight maintenance process in Algorithm 5. It implies that implicit weight maintenance process is sublinear in $m$.

**Lemma 6.5** (Step 3, implicit weight maintenance. Informal version of Lemma F.4). *The implicit weight maintenance of Algorithm 1(its formal version is Algorithm 5) runs in time $o(nm) \cdot (d + \log m)$.*

## 7 LOWER BOUND

In this section, we provide a lower bound discussion here.

### 7.1 PREVIOUS RESULTS ON MAXIMUM INNER PRODUCT

We state a result considering the maximum bichromatic inner product lower bound here Chen (2018). For more background about complexity involving SETH and OVC, we refer reader to Appendix A.10

**Definition 7.1** (Bichromatic Maximum Inner Product ($\mathsf{Max-IP}_{n,d}$))**.** *For $n, d \in \mathcal{N}$, the $\mathsf{Max-IP}_{\mathsf{n,d}}$ probelm is defined as: given two sets $A, B$ each consisting of $n$ vectors from $\{0,1\}^d$ compute $\mathsf{OPT}(A, B) := \max_{a \in A, b \in B} a \cdot b$. We use $\mathbb{Z} - \mathsf{Max-IP}_{n,d}$ to denote the same problem, with $A, B$ being sets of vectors from $\mathbb{Z}^d$.*

**Theorem 7.2** (Maximum bichromatic inner product lower bound Chen (2018))**.** *Under assumption of SETH (Definition A.23) or OVC (Definition A.25), there is a constant $c$ such that any exact algorithm for $\mathbb{Z} - \mathsf{Max-IP}_{\mathsf{n,d}}$ in dimension $d = c^{\log^* n}$ requires $n^{2-o(1)}$ time, with vectors of $O(\log n)$-bit entries.*

### 7.2 Lower Bound by Reduction from Maximum Inner Product Search

Here we provide the theorem for the lower bound.

**Theorem 7.3** (Lower Bound for DSWZ, formal version of Theorem 1.4)**.** *Let $c \in (0, 1)$ be a constant. Let $d = 2^{O(\log^* n)}$. Assume $m = \mathrm{poly}(n)$. Given SETH and OVC, the DSWZ (Definition 1.3) cannot have update time of $O(n^{1-\epsilon})$ and query time of $O(m^{1-c}n^{c-\epsilon})$ for any $\epsilon \in (0, c)$.*

*Proof.* Let $\epsilon > 0$ be a parameter. We consider $m \geq n$ and $d = c^{\log^* n}$, where $c$ is defined as in Theorem 7.2. Assume there is a data structure on a instance on $m$ weights and $n$ data points in $d + 1$ dimensional space, with an update time of $O(n^{1-\epsilon})$ and query time of $O(n^{1-\epsilon})$. Following Theorem 7.2, we construct a hard instance of $\mathbb{Z} - \mathsf{Max-IP}_{m,d}$ problem with $W = \{w_1, \ldots, w_m\} \subseteq \mathbb{Z}^d, X = \{x_1, \ldots, x_m\} \subseteq \mathbb{Z}^d$.

We first construct two new data sets $\overline{W}, \overline{X} \subseteq \mathbb{Z}^{d+1}$ where for each $w_i \in W$, we let $(\overline{w_i})_{d+1}$ to be set later, and for every $x_i \in X$, we set $(\overline{x_i})_{d+1} = -1$. And to utilize DSWZ, we divide $\overline{X}$ into $T = O(m/n)$ different sets, we use $\overline{X}_t$ to denote $t$-th set for all $t \in [T]$. Each set $\overline{X}^{(t)}$ has size of $n$.

Now we utilize DSWZ data structure to solve it. We apply our DSWZ for each pair $(\overline{W}, \overline{X}^{(t)})$ for every $t \in [T]$. The general idea is to perform a binary search on the value of $\mathsf{OPT}(W, X)$, with calling to DSWZ in each iteration. Notice that, the number of iterations is at most $O(\log n)$.

Assume in some iteration, the threshold is $s \in \mathbb{Z}$. We call $\textsc{Update}(s, j)$ to update each $\overline{w}_j^{(t)}$ by setting the $d + 1$-th entry to be $s$ for every $j \in [m]$ and every $t \in [T]$. This takes time $O(m \cdot n^{1-\epsilon})$ for each $t \in [T]$, by the assumption of running time. Now for each $i \in [n]$, we call $\textsc{Query}(i, 0)$. By the construction of our datasets, we know that $\langle \overline{w}_i, \overline{x}_j \rangle \geq 0$ iff $\langle w_i, x_j \rangle \geq s$. This step takes time $O(n \cdot m^{1-c}n^{c-\epsilon}) = O(m^{1-c}n^{1+c-\epsilon})$ time and outputs all the pair of $(i, j)$ such that $\langle w_i, x_j \rangle \geq s$ for each $t \in [T]$. Combining the above results, we have the total running time to solve $\mathbb{Z} - \mathsf{Max-IP}_{n,d}$ being

$$O(T \cdot (m \cdot n^{1-\epsilon} + m^{1-c}n^{1+c-\epsilon})) = O(m^2 n^{-\epsilon} + m^{2-c}n^{c-\epsilon}),$$

which can be bounded by $O(m^{2-\widetilde{\epsilon}})$ since $m = \mathrm{poly}(n)$ for some $\widetilde{\epsilon} < \epsilon$.

Above two steps implies that each iteration of the binary search takes time $O(m^{2-\widetilde{\epsilon}})$. Thus, $\mathbb{Z} - \mathsf{Max-IP}_{m,d}$ problem can be solved in time $O(m^{2-\widetilde{\epsilon}} \cdot \log n) = O(m^{2-o(1)})$. This contradicts Theorem 7.2. Hence, this data structure cannot exist. $\qquad\square$

## 8 Conclusion

The computational cost of training massively overparametrized DNNs is posing a major scalability barrier to the progress of AI, and motivates rethinking the traditional SGD-based training algorithms. For a neural network with $m$ parameters and an input batch of $n$ datapoints in $\mathbb{R}^d$, previous state-of-art Brand et al. (2021); Zhang et al. (2019) show that dramatically fewer iterations (epochs) $T_\epsilon$ can be achieved via second-order methods, albeit with $O(mnd + n^3)$ cost per iteration, i.e., $O(T_\epsilon \cdot mnd)$ overall time to reduce training error below $\epsilon$. Our work proposes a simple yet powerful view of the gradient flow process on wide DNNs ($m = \mathrm{poly}(n)$), as a collection of *slowly-changing binary search trees*, enabling the design of a training algorithm for 2-layer overparametrized DNNs in *sublinear* cost-per-iteration, while enjoying the ultra-fast convergence rate of second-order (Gauss-Newton) methods, i.e., in total time $\widetilde{O}(T_\epsilon + mnd)$ instead of the aforementioned $\widetilde{O}(T_\epsilon \cdot mnd)$.

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

APPENDIX

**Roadmap.** The appendix of this paper is organized as follows. Section A presents the preliminary tools which are used in the other parts of appendix. Section B presents the complete description and implementation of the threshold search data structure. Section C presents the formal algorithm representation of our fast neural network training algorithm. Section D shows the omitted proofs of some lemmas in the convergence analysis. Section E presents the complete the induction hypothesis proof to show the convergence of our training algorithm. Section F presents the formal proof of the running time of our training algorithm, especially shows a more specific conclusion compared with the main body. Section G presents a formal analysis of the applying conditions of our training algorithm.

## A  PRELIMINARY

This section shows some preliminary tools to be used later. We start with defining some basic notations in Section A.1. In Section A.2 we provide more concept for the model formalization. In Section A.3 we describe neural tangent kernel and its relation with data separability. In Section A.4 we introduce a sketching tool. In Section A.5 we introduce the result for fast matrix multiplication. In Section A.6 we state the probability tools to be used. In Section A.7 we presented a previous result on the relationship of changes of weights and the change of the shifted NTK matrix. In Section A.8 we provide some useful results about fast regression. In Section A.9 we provide results about sparsity-based preserving. We introduce SETH and OVC from computational complexity in Section A.10.

### A.1  NOTATIONS.

For any positive integer $n$, we use $[n]$ to denote set $\{1, 2, \cdots, n\}$. For any function $f$, we use $\widetilde{O}(f)$ to denote $f \cdot \text{poly}(\log f)$. For two vectors $w$ and $x$, we use $\langle w, x \rangle$ to denote inner product. We use $a^\top$ to denote the transpose of $a$. We use $\mathbb{E}[]$ to denote expectation and $\Pr[]$ for probability. For convenience, we use FMM to denote fast matrix multiplication. We use NTK to denote neural tangent kernel. We use ReLU to denote rectified linear unit. We use NN to denote neural network. We use CPI to represent the cost per iteration. We use PSD to denote positive semidefinite. We use $\log^* n$ to denote the iterated logarithm, which is a function grows slowly as barely larger than a constant.

### A.2  MORE ABOUT THE MODEL

**Definition A.1** (Loss function). *Suppose the dimension of input is $d$, the number of intermediate nodes (or hidden neurons) is $m$, the dimension of output is $1$, the batch size is $n$ and the shifted parameter is $b$ ($b \geq 0$). For a fixed set of $n$ points $x_1, x_2, \cdots, x_n \in \mathbb{R}^d$ and their corresponding labels $y_1, y_2, \cdots, y_n \in \mathbb{R}$. Consider the following loss function:*

$$\mathcal{L}(W) := \frac{1}{2} \sum_{i=1}^{n} (y_i - f(W, x_i, a))^2.$$

By mathematics,

$$\frac{\partial f(W, x, a)}{\partial w_r} = \frac{1}{\sqrt{m}} a_r x \mathbf{1}_{w_r^\top x \geq b}, \quad \forall r \in [m]. \tag{1}$$

Thus one can calculate the gradient of loss function $\mathcal{L}$

$$\frac{\partial \mathcal{L}(W)}{\partial w_r} = \frac{1}{\sqrt{m}} \sum_{i=1}^{n} (f(W, x_i, a) - y_i) \cdot a_r \cdot x_i \cdot \mathbf{1}_{w_r^\top x_i \geq b}. \tag{2}$$

Then we define the prediction function, the Jacobi matrix, and the Gram matrix.

**Definition A.2** (prediction function). *For a batch of inputs $\{(x_i, y_i)\}_{i \in [n]} \in \mathbb{R}^d \times \mathbb{R}$, we denote*

$$\alpha_{r,i}(t) := \phi(\langle w_r(t), x_i \rangle)$$

*for every $r \in [m]$ and $i \in [n]$.*

*Prediction function $f_t : \mathbb{R}^{d \times n} \to \mathbb{R}^n$ at time $t$ is defined as follows*

$$f_t = \frac{1}{\sqrt{m}} \sum_{r \in [m]} \begin{bmatrix} a_r \cdot \alpha_{r,1}(t) \\ a_r \cdot \alpha_{r,2}(t) \\ \vdots \\ a_r \cdot \alpha_{r,n}(t) \end{bmatrix}$$

*Note that $w_r(t)$ is the $r$-th weight of the first layer after training of $t$ times.*

*For convenience, we define weight matrix*

$$W_t = [w_1(t)\ w_2(t)\ \cdots\ w_m(t)] \in \mathbb{R}^{d \times m}$$

*In addition, we write data matrix*

$$X = [x_1\ x_2\ \cdots\ x_n] \in \mathbb{R}^{d \times n}.$$

**Definition A.3** (Jacobi matrix and related definitions)**.** *For each $i \in [n]$, $r \in [m]$ and $t \in [T]$, we define*

$$\beta_{r,i}(t) := \mathbf{1}_{\langle w_r(t), x_i \rangle \geq b}.$$

*For every time step $t$, we use $J_t \in \mathbb{R}^{n \times m}$ to denote the Jacobian matrix at $t$. Formally, it can be written as*

$$J_t = \frac{1}{\sqrt{m}} \begin{bmatrix} a_1 x_1^\top \beta_{1,1}(t) & a_2 x_1^\top \beta_{2,1}(t) & \cdots & a_m x_1^\top \beta_{m,1}(t) \\ a_1 x_2^\top \beta_{1,2}(t) & a_2 x_2^\top \beta_{2,2}(t) & \cdots & a_m x_2^\top \beta_{m,2}(t) \\ \vdots & \vdots & \ddots & \vdots \\ a_1 x_n^\top \beta_{1,n}(t) & a_2 x_n^\top \beta_{2,n}(t) & \cdots & a_m x_n^\top \beta_{m,n}(t) \end{bmatrix}.$$

*For each $i \in [n]$, we define $J_t(x_i)$ as the $i$-th row of $J_t$.*

**Definition A.4** (Gram matrix)**.** *Let $G_t \in \mathbb{R}^{n \times n}$ denote the Gram matrix. Then $G_t$ can be formally written as $G_t = J_t J_t^\top$. The $(i,j)$-th entry of $G_t$ is the inner product between gradient in terms of $x_i$ and the gradient in terms of $x_j$, i.e.,*

$$(G_t)_{i,j} := \langle \frac{f(W_t, x_i)}{\partial W}, \frac{f(W_t, x_j)}{\partial W} \rangle.$$

Jacot et al. (2018); Du et al. (2019); Song et al. (2021a) gave a crucial observation that the asymptotic of the Gram matrix $G$ is equal to a PSD matrix $K \in \mathbb{R}^{n \times n}$. The formal definition is

$$K(x_i, x_j) := \mathop{\mathbb{E}}_{w \sim \mathcal{N}(0, I)} \left[ x_i^\top x_j \mathbf{1}_{\langle w, x_i \rangle \geq b, \langle w, x_j \rangle \geq b} \right]. \tag{3}$$

Jacot et al. (2018); Du et al. (2019) only consider the case where $b = 0$ and Song et al. (2021a) consider the general case $b \geq 0$.

**Remark A.5.** *We use $\lambda$ to denote the minimal eigenvalue of the kernel matrix $K$ defined in Eq. (3).*

### A.3 NEURAL TANGENT KERNEL AND ITS RELATION WITH DATA SEPARABILITY

Neural Tangent Kernel (NTK) is a Kernel matrix related to a multi-layer ReLU activated neural network. It is crucial in the analysis of Jacobi matrix. Song et al. (2021a) expanded the related concepts and revealed their properties, especially its relation to the data separability of an input batch.

As for data separability, it is a common assumption to the input of a neural network, and it has been used in many over-parameterized neural network literature Li & Liang (2018); Allen-Zhu et al. (2019b). We first define kernels,

**Definition A.6.** *Let $b \geq 0$ be the shift parameter. We define continuous version of the shifted NTK $H^{\mathrm{cts}}$ and discrete version of shifted NTK $H^{\mathrm{dis}}$ as*

$$H^{\mathrm{cts}}_{i,j} := \mathop{\mathbb{E}}_{w \sim \mathcal{N}(0, I)} [x_i^\top x_j \mathbf{1}_{w^\top x_i \geq b, w^\top x_j \geq b}],$$

$$H^{\mathrm{dis}}_{i,j} := \frac{1}{m} \sum_{r=1}^{m} [x_i^\top x_j \mathbf{1}_{w_r^\top x_i \geq b, w_r^\top x_j \geq b}].$$

Next, we define data separability,

**Definition A.7** (Separability of input data)**.** *Suppose we are given $n$ (normalized) input data points*

$$\{x_1, x_2, \cdots, x_n\} \subseteq \mathbb{R}^d.$$

*Assume those points satisfy that $\forall i \in [n], \|x_i\|_2 = 1$. For each $i, j$, we define*

$$\delta_{i,j}^+ = x_i + x_j \text{ and } \delta_{i,j}^- = x_i - x_j.$$

*Let $\delta$ be the data separability parameter, formally,*

$$\delta := \min_{i \neq j}\{\min\{\|\delta_{i,j}^+\|_2, \|\delta_{i,j}^-\|_2\}\}.$$

Song et al. (2021a) has given a property of the minimal eigenvalue of the NTK of a shifted ReLU activated neural network.

**Lemma A.8** (Lemma C.1 in Song et al. (2021a))**.** *Let $m$ be the number of samples of $H^{\mathrm{dis}}$. As long as*

$$m = \Omega(\lambda^{-1} n \log(n/\rho)),$$

*then*

$$\Pr[\lambda_{\min}(H^{\mathrm{dis}}) \geq \frac{3}{4}\lambda] \geq 1 - \rho.$$

Prior work (Oymak & Soltanolkotabi (2020)) has shown the relation between the data separability of the input of a neural network and the eigenvalue of the Kernel. But their work focuses on unshifted ReLU activated neural network. For shifted ReLU activated neural network, Song et al. (2021a) provided a further generalization to the shifted Kernel matrix.

**Theorem A.9** (Theorem F.1 in Song et al. (2021a))**.** *Consider $n$ points $x_1, \ldots, x_n \in \mathbb{R}^d$ with $\ell_2$-norm all equal to 1, and consider a random variable $w \sim \mathcal{N}(0, I_d)$. Define matrix*

$$X \in \mathbb{R}^{n \times d} = [x_1 \ \ldots \ x_n]^\top.$$

*Suppose the data separability of the $n$ points is $\delta$ where $\delta < \sqrt{2}$. Let shift parameter $b \geq 0$. Recall the continuous Hessian matrix $H^{\mathrm{cts}}$ is defined by*

$$H_{i,j}^{\mathrm{cts}} := \mathop{\mathbb{E}}_{w \sim \mathcal{N}(0,I)}[x_i^\top x_j \mathbf{1}_{w^\top x_i \geq b, w^\top x_j \geq b}], \forall(i,j) \in [n] \times [n].$$

*Let $\lambda := \lambda_{\min}(H^{\mathrm{cts}})$. Then $\lambda$ has the follow sandwich bound,*

$$\lambda \in [\exp(-b^2/2) \cdot \frac{\delta}{100n^2}, \exp(-b^2/2)].$$

### A.4 A SKETCHING TOOL

Sarlós Sarlos (2006) firstly introduced the notation of subspace embedding. Many numerical linear algebra applications have used that concept and its variations Clarkson & Woodruff (2013); Nelson & Nguyên (2013); Razenshteyn et al. (2016); Song et al. (2017; 2019); Song & Yu (2021). The formal definition is:

**Definition A.10** (Oblivious subspace embedding, OSE Sarlos (2006))**.** *Given an $N \times k$ matrix $B$, an $(1 \pm \epsilon)$ $\ell_2$-subspace embedding for the column space of $B$ is a matrix $S$, such that for any $x \in \mathbb{R}^k$,*

$$(1 - \epsilon)\|Bx\|_2^2 \leq \|SBx\|_2^2 \leq (1 + \epsilon)\|Bx\|_2^2.$$

*Equivalently, let $U$ be the matrix whose columns form an orthonormal basis containing the column vectors of $B$, then*

$$\|I - U^\top S^\top S U\|_2 \leq \epsilon.$$

It is known that subspace embedding can be given by a Fast-JL sketching matrix Ailon & Chazelle (2006); Drineas et al. (2006); Tropp (2011); Drineas et al. (2012); Lu et al. (2013); Price et al. (2017) with a classical $\epsilon$-net argument,

**Lemma A.11.** *Assume that $N = \mathrm{poly}(k)$. Assume $\delta \in (0, 0.1)$. For a matrix $B \in \mathbb{R}^{N \times k}$, we can produce an $(1 \pm \epsilon)$ $\ell_2$-subspace embedding $S \in \mathbb{R}^{k \, \mathrm{poly}(\log(k/\delta))/\epsilon^2 \times N}$ for $B$ with probability at least $1 - \delta$.*

*In addition, $SB$ takes $O(Nk \cdot \mathrm{poly} \log k)$ time to be generated.*

### A.5 FAST MATRIX MULTIPLICATION

We state a standard fact for fast matrix multiplication (FMM).

**Fact A.12** (FMM). *Given an $n \times n$ matrix $A$ and another $n \times n$ matrix $B$, the time of multiplying $A$ and $B$ is $n^{\omega}$, where $\omega \approx 2.373$ is the exponent of matrix multiplication. Currently, $\omega \approx 2.373$ Williams (2012).*

### A.6 PROBABILITY TOOLS

We list some probability tools which are useful in our analysis.

**Lemma A.13** (Chernoff bound Chernoff (1952)). *Let $Z = \sum_{i=1}^{n} Z_i$, where $Z_i = 1$ with probability $p_i$ and $Z_i = 0$ with probability $1 - p_i$, and all $Z_i$ are independent. We define $\mu = \mathbb{E}[Z] = \sum_{i=1}^{n} p_i$. Then*
*1. $\Pr[Z \geq (1 + \delta)\mu] \leq \exp(-\delta^2 \mu/3)$, $\forall \delta > 0$ ;*
*2. $\Pr[Z \leq (1 - \delta)\mu] \leq \exp(-\delta^2 \mu/2)$, $\forall \delta \in (0, 1)$.*

**Lemma A.14** (Hoeffding bound Hoeffding (1963)). *Let $Z_1, \cdots, Z_n$ denote $n$ independent bounded variables in $[a_i, b_i]$. Let $c_i = (b_i - a_i)$ Let $Z = \sum_{i=1}^{n} Z_i$, then we have*

$$\Pr[|Z - \mathbb{E}[Z]| \geq t] \leq 2 \exp\left(-\frac{2t^2}{\sum_{i=1}^{n} c_i^2}\right).$$

**Lemma A.15** (Anti-concentration inequality). *Let $Z \sim \mathcal{N}(0, \sigma^2)$, that is, the probability density function of $Z$ is given by $\phi(x) = \frac{1}{\sqrt{2\pi\sigma^2}} e^{-\frac{x^2}{2\sigma^2}}$. Then*

$$\Pr[|Z| \leq t] \leq \frac{4}{5}\frac{t}{\sigma}.$$

### A.7 PERTURBED $w$ FOR SHIFTED NTK

We present a lemma from previous work in Song et al. (2021a). They show that in general, small changes of weights only lead to small change of the Shifted NTK matrix.

**Lemma A.16** (Lemma C.2 in Song et al. (2021a), perturbed $w$ for shifted NTK). *Suppose $b > 0$. Assume $R \leq 1/b$. Suppose $m = \Omega(\lambda^{-1} n \log(n/\rho))$. Define function $H$ which maps $\mathbb{R}^{m \times d}$ to $\mathbb{R}^{n \times n}$ as follows:*

$$\text{the } (i, j)\text{-th entry of } H(W) \text{ is } \frac{1}{m} x_i^\top x_j \sum_{r=1}^{m} \mathbf{1}_{w_r^\top x_i \geq b, w_r^\top x_j \geq b}.$$

*Let $m$ vectors $w_1, w_2, \cdots, w_m$ sampled from $\mathcal{N}(0, I_d)$ and let $\widetilde{W} = [w_1 \ w_2 \ \cdots \ w_m]$. Then there exist constants $c > 0$ and $c' > 0$ such that, for all $W \in \mathbb{R}^{d \times m}$ with $\|\widetilde{W} - W\|_{\infty,2} \leq R$, the following holds:*

- *Part 1, $\|H(\widetilde{W}) - H(W)\|_F \leq n \cdot \min\{c \cdot \exp(-b^2/2), 3R\}$ holds with prob. $\geq 1 - n^2 \cdot \exp(-m \cdot \min\{c' \cdot \exp(-b^2/2), R/10\})$.*

- *Part 2, $\lambda_{\min}(H(W)) \geq \frac{3}{4}\lambda - n \cdot \min\{c \cdot \exp(-b^2/2), 3R\}$ holds with prob. $\geq 1 - n^2 \cdot \exp(-m \cdot \min\{c' \cdot \exp(-b^2/2), R/10\}) - \rho$.*

### A.8 FAST REGRESSION SOLVER

We list some useful conclusions about fast regression from Brand et al. (2021).

**Lemma A.17** (Lemma B.2 in Brand et al. (2021)). *Consider the the regression problem*

$$\min_x \|Bx - y\|_2^2.$$

*Suppose $B$ is a PSD matrix with $\frac{3}{4} \leq \|Bx\|_2 \leq \frac{5}{4}$ holds for all $\|x\|_2 = 1$. Using gradient descent, after $t$ iterations, we obtain*

$$\|B(x_t - x^\star)\|_2 \leq c^t \cdot \|B(x_0 - x^\star)\|_2$$

*for some constant $c \in (0, 0.9]$.*

**Lemma A.18** (Lemma B.1 in Brand et al. (2021)). *Suppose there is a matrix $Q \in \mathbb{R}^{N \times k}$ ($N \geq k \operatorname{poly}(\log k)$), with condition number $\kappa$ (i.e., $\kappa = \sigma_{\max}(Q)/\sigma_{\min}(Q)$), consider this minimization problem*

$$\min_{x \in \mathbb{R}^k} \|Q^\top Q x - y\|_2. \tag{4}$$

*It is able to find a vector $x'$*

$$\|Q^\top Q x' - y\|_2 \leq \|y\|_2 \cdot \epsilon$$

*in $\mathcal{T}_{\mathrm{precond}} + \mathcal{T}_{\mathrm{iters}} \cdot \mathcal{T}_{\mathrm{cost}}$ time where*

- *$\mathcal{T}_{\mathrm{precond}} = \widetilde{O}(Nk + k^3)$ without using FMM, $\widetilde{O}(Nk + k^\omega)$ using FMM.*

- *$\mathcal{T}_{\mathrm{iters}} = O(\log(\kappa/\epsilon))$,*

- *$\mathcal{T}_{\mathrm{cost}} = \widetilde{O}(Nk)$.*

The above lemma and preconditioning property implies that the iterative regression will take $\log(\kappa/\epsilon)$ iterations.

**Corollary A.19.** *Solving regression problem (4) needs $O(\log(\kappa/\epsilon))$ iterations using the above method.*

The cost per iteration in the iterative regression is too slow for our application. In Section F, we will show how to improve the cost per iteration while maintaining the same number of iterations.

### A.9 Sparsity-based Preserving

We present a tool from the paper Song et al. (2021a). Firstly, we provide a definition.

**Definition A.20.** *For every $t \in \{0, 1, \cdots, T\}$. For every $i \in [n]$. We use $\mathcal{S}_{i,\mathrm{fire}}(t) \subset [m]$ to represent the set of neurons that are "fire" at time $t$, i.e.,*

$$\mathcal{S}_{i,\mathrm{fire}}(t) := \{r \in [m] : \langle w_r(t), x_i \rangle > b\}.$$

*For all $t \in \{0, 1, \cdots, T\}$, define $k_{i,t} := |\mathcal{S}_{i,\mathrm{fire}}(t)|$ to express the number of fire neurons for $x_i$.*

The following lemma (Lemma 3.8 in Song et al. (2021a)) show that with the increase of the shifted paramater, the initial neural network will become sparser.

**Lemma A.21** (Sparsity preserving). *Assume $m$ is number of neurons. For shifted parameter $b > 0$, if we use $\phi_b$ as the activation function of a 2-layer neural network, then after initialization, with prob. $\geq 1 - n \cdot \exp(-\Omega(m \cdot \exp(-b^2/2)))$, we have for every $i$, $k_{i,0}$ is not larger than $O(m \cdot \exp(-b^2/2))$.*

Using the above lemma, we can obtain the following result,

**Corollary A.22.** *If we set shifted parameter $b = \sqrt{0.48 \log m}$ then $k_0 = m^{0.76}$. For $t = m^{0.76}$,*

$$\Pr\left[|\mathcal{S}_{i,\mathrm{fire}}(0)| > 2m^{0.76}\right] \leq \exp\left(-\min\{mR, O(m^{0.76})\}\right).$$

### A.10 SETH and OVC

Here we introduce some notions from computational complexity, for our analysis of lower bound.

**Definition A.23** (Strong exponential time hypothesis (SETH) Impagliazzo & Paturi (2001); Calabro et al. (2009)). *For any $\varepsilon > 0$, there exists a $k = k(\varepsilon)$ such that the $k$-SAT problem with $n$ variables cannot be solved in time $O(2^{1-\epsilon}n)$.*

In order to introduce OVC, we need to define Orthogonal Vector problem first.

**Definition A.24** (Orthogonal Vector problem). *Given a set of $n$ vectors $\{v_1, \ldots, v_n\} \subseteq \{0, 1\}^d$ in $d$-dimensional space. We ask if there exists $(i, j) \in [n] \times [n]$ such that $\langle v_i, v_j \rangle = 0$.*

**Definition A.25** (Orthogonal vector conjecture (OVC) Williams (2005); Abboud et al. (2014b); Backurs & Indyk (2016); Abboud et al. (2015)). *For every $\varepsilon > 0$, there exists a $c = c(\varepsilon) > 1$ such that OV cannot be solved in $n^{2-\varepsilon}$ time when $d = c \log n$.*

## B    THRESHOLD SEARCH DATA STRUCTURE

This section gives a data structure which can efficiently find all the weights $w_j$ such that $\langle w_j, x_i \rangle \geq \tau$ for each given input $x_i$ and real number $\tau$. Specifically, Section B.1 formally proposes this data structure. Section B.2 proves the running time of INIT satisfies the requirement of Theorem B.1. Section B.3 proves the running time of UPDATE satisfies the requirement of Theorem B.1. Section B.4 proves the running time of QUERY satisfies the requirement of Theorem B.1. Section B.5 proves the correctness of QUERY in Theorem B.1.

### B.1    MAIN RESULT

In this section, we are going to present our key theorem (Theorem B.1).

**Theorem B.1** (Our tree data structure). *There exists a data structure which requires $O(mn+nd+md)$ spaces and supports the following procedures:*

- INIT($\{w_1, w_2, \cdots, w_m\} \subset \mathbb{R}^d, \{x_1, x_2, \cdots, x_n\} \subset \mathbb{R}^d$. *Given a series of weights $w_1, w_2, \cdots, w_m$ and datas $x_1, x_2, \cdots, x_n$, it preprocesses in time $O(mnd)$.*

- UPDATE($z \in \mathbb{R}^d, j \in [m]$). *Given a new weight vector $z \in \mathbb{R}^d$ and index $j \in [m]$, it updates weight $w_j$ with $z$ in time $O(n(d + \log m))$.*

- QUERY($i \in [n], \tau \in \mathbb{R}$). *Given a query index $i \in [n]$ and a threshold $\tau \in \mathbb{R}$, it finds all index $j \in [m]$ such that $\langle w_j, x_i \rangle \geq \tau$ in time $O(K_q \cdot \log m)$, where $K_q := |\{j \in [m] \mid \langle w_j, x_i \rangle \geq \tau\}|$.*

*Proof.* Since $W$ takes $O(md)$ space, $X$ takes $O(nd)$ space, each binary tree $T_i$ stores $O(m)$ data, the data structure uses $O(mn + nd + md)$. Then we use the following Lemma B.2, B.3, B.4 and B.5 to prove the correctness and running time of this data structure. □

---

**Algorithm 2** Our tree data structure: members, init

```
 1: data structure TREE                                                          ▷ Theorem B.1
 2: members
 3:     W ∈ ℝ^{m×d} (m weight vectors)
 4:     X ∈ ℝ^{n×d} (n data points)
 5:     Binary tree T₁, T₂, ⋯, Tₙ   ▷ We create n binary search trees, each tree uses O(mn) space
 6: end members
 7:
 8: public:
 9: procedure INIT(w₁, w₂, ⋯, wₘ ∈ ℝ^d, x₁, x₂, ⋯, xₙ ∈ ℝ^d)                    ▷ Lemma B.2
10:     for i = 1 → n do
11:         xᵢ ← xᵢ
12:     end for
13:     for j = 1 → m do
14:         wⱼ ← wⱼ
15:     end for
16:     for i = 1 → n do                                           ▷ for data point, we create a tree
17:         for j = 1 → m do
18:             uⱼ ← ⟨xᵢ, wⱼ⟩
19:         end for
20:         Tᵢ ← MAKEBINARYSEARCH(u₁, ⋯, uₘ)
21:                                          ▷ Each node stores the maximum value for his two children
22:     end for
23: end procedure
24: end data structure
```

---

---

**Algorithm 3** Our dynamic data structure: update

1: **data structure** TREE                 ▷ Theorem B.1
2: **public:**
3: **procedure** UPDATE($z \in \mathbb{R}^d, j \in [m]$)         ▷ Lemma B.3
4:   $w_j \leftarrow z$
5:   **for** $i \in [n]$ **do**
6:     $l \leftarrow$ the $j$-th leaf of tree $T_i$
7:     $l$.value $\leftarrow \langle z, x_i \rangle$
8:     **while** $l$ is not root **do**
9:       $p \leftarrow$ parent of $l$
10:      $a \leftarrow$ left child of $p$
11:      $b \leftarrow$ right child of $p$
12:      $p$.value $\leftarrow \max\{a.\text{value}, b.\text{value}\}$
13:      $l \leftarrow p$
14:     **end while**
15:   **end for**
16: **end procedure**
17: **end data structure**

---

**Algorithm 4** Our dynamic data structure: query

1: **data structure** TREE                 ▷ Theorem B.1
2: **public:**
3: **procedure** QUERY($i \in [n], \tau \in \mathbb{R}_{\geq 0}$)         ▷ Lemma B.4
4:   QRECURSIVE($\tau, \text{root}(T_i)$)
5: **end procedure**
6:
7: **private:**
8: **procedure** QRECURSIVE($\tau \in \mathbb{R}_{\geq 0}, r \in T$)
9:   **if** $r$ is leaf **then**
10:    **if** $r$.value $> \tau$ **then**
11:     **return** $r$.index
12:    **end if**
13:   **else**
14:    $r_1 \leftarrow$ left child of $r$, $r_2 \leftarrow$ right child of $r$
15:    **if** $r_1$.value $\geq \tau$ **then**
16:     $S_1 \leftarrow$ QRECURSIVE($\tau, r_1$)
17:    **end if**
18:    **if** $r_2$.value $\geq \tau$ **then**
19:     $S_2 \leftarrow$ QRECURSIVE($\tau, r_2$)
20:    **end if**
21:   **end if**
22:   **return** $S_1 \cup S_2$
23: **end procedure**
24: **end data structure**

---

### B.2 RUNNING TIME OF INIT

We prove Lemma B.2, which presents the running time for the INIT operation. The corresponding algorithm is shown in Algorithm 2.

**Lemma B.2** (Running time of INIT)**.** *Given a series of weights $\{w_1, w_2, \cdots, w_m\} \subset \mathbb{R}^d$ and datas $\{x_1, x_2, \cdots, x_n\} \subset \mathbb{R}^d$, the procedure* INIT *(Algorithm 2) preprocesses in time $O(nmd)$.*

*Proof.* The INIT consists of two independent for loops and two recursive for loopss. The first for loop (start from line 10) has $n$ iterations, which takes $O(nd)$ time. The second for loop (start from line 13) has $m$ iterations, which takes $O(md)$ time. Now we consider the recursive for loop. The outer loop (line 16) has $n$ iterations. In inner loop has $m$ iterations. In every iteration of the inner loop, line 18 runs in $O(d)$ time. Line 20 takes $O(m)$ time. Putting it all together, the INIT runs in

time

$$O(nd + md + n(md + m))$$
$$= O(nmd)$$

So far, the proof is finished. $\square$

### B.3 RUNNING TIME OF UPDATE

We prove Lemma B.3. The corresponding algorithm is shown in Algorithm 3.

**Lemma B.3** (Running time of UPDATE). *Given a weight $z \in \mathbb{R}^d$ and index $j \in [m]$, the procedure* UPDATE *(Algorithm 3) updates weight $w_j$ with $z$ in $O(n \cdot (d + \log m))$ time.*

*Proof.* The time of UPDATE mainly comes from the forloop (line 5), which consists of $n$ iterations. In each iteration, line 7 takes $O(d)$ time, and the while loop takes $O(\log m)$ time since it go through a path bottom up. Putting it together, the running time of UPDATE is $O(n(d + \log m))$. $\square$

### B.4 RUNNING TIME OF QUERY

We prove Lemma B.4, which is the running time for the QUERY operation. The corresponding algorithm is shown in Algorithm 4.

**Lemma B.4** (Running time of QUERY). *Given a query index $i \in [n]$ and a threshold $\tau > 0$, the procedure* QUERY *(Algorithm 4) runs in time $O(K_q \cdot \log m)$, where $K_q := |\{j \in [m] : \langle w_j, x_i \rangle > \tau\}|$.*

*Proof.* The running time comes from QRECURSIVE with input $\tau$ and $\mathrm{root}(T_i)$. In QRECURSIVE, we start from the root node $r$ and find indices in a recursive way. The INIT guarantees that for a node $r$ satisfying $r.\mathrm{value} > \tau$, the sub-tree with root $r$ must contains a leaf whose value is greater than $\tau$ If not satisfied, all the values of the nodes in the sub-tree with root $r$ is less than $\tau$. This guarantees that all the paths it searches do not have any branch that leads to unnecessary leaves. Our data structure will report all the indices $i$ satisfying $\langle w_i, q \rangle > \tau$. Since the depth of $T$ is $O(\log m)$, the running time of QUERY is $O|K_q| \cdot \log m)$. $\square$

### B.5 CORRECTNESS OF QUERY

We prove Lemma B.5, which shows the correctness for the QUERY operation.

**Lemma B.5** (Correctness of QUERY). *Given a query index $i \in [n]$ and a threshold $\tau > 0$, the procedure* QUERY *(Algorithm 4) finds all index $j \in [m]$ such that $\langle x_i, w_j \rangle > \tau$.*

*Proof.* Fix $i \in [n]$, for all $j \in [m]$, suppose the $j$-th leaf of $T_i$ is $l$, the root of $T_i$ is $r$, and the path from $r$ to $l$ is

$$r = p_0 \to p_1 \to \cdots \to p_k = l.$$

If $\langle x_i, w_j \rangle > \tau$, first $j \in \mathrm{QRECURSIVE}(p_k)$, then, suppose $j \in \mathrm{QRECURSIVE}(p_{t+1})$, then $p_{t+1}.\mathrm{value} \geq \langle w_j, x_i \rangle > \tau$, thus $j \in \mathrm{QRECURSIVE}(p_{t+1}) \subseteq \mathrm{QRECURSIVE}(p_t)$. Hence by induction, $j \in \mathrm{QRECURSIVE}(p_0) = \mathrm{QUERY}(i, \tau)$. If $\langle x_i, w_j \rangle \leq \tau$, since $l.\mathrm{value} \geq \tau$, $j$ will not be returned. Thus QUERY finds exactly all the index $j \in [m]$ such that $\langle x_i, w_j \rangle > \tau$.

$\square$

## C FORMAL ALGORITHM REPRESENTATION

We have given a concise representation of our training algorithm (Algorithm 1) in previous sections, for facilitating understanding. For the sake of completeness and convenient implementation, this section gives a formal algorithm representation of our fast neural network training algorithm. (See Algorithm 5.)

This algorithm starts with initializing weights $W_0$ and setting shifted parameter $b$. After that, it repeatedly executes sketch computing, iterative regression and implicit weight maintenance until enough times. Specifically, sketch computing computes a sketch matrix $S$ for $J_t^\top$ with property $\|SJ_t^\top x\|$ is closed to $\|J_t^\top x\|$ for every $x$ with large probability. Iterative regression makes use of a fast regression solver to find an approximate solution of

$$g_t := \arg\min_g \|J_t J_t^\top g - (f_t - y)\|$$

with the help of the sketch matrix $S$.

Implicit weight maintenance utilizes the threshold search data structure to update weights using the information propagated by the iterative regression.

# D  MORE DETAILS ABOUT CONVERGENCE ANALYSIS

The convergence analysis is shown in Section 5. It uses Lemma 5.2, Lemma 5.3 and Lemma 5.4 without proofs. In this section, we formally present the proofs of the three lemmas. In Section D.1, we provide the proof of Lemma 5.2. In Section D.2, we provide the proof of Lemma 5.3. In Section D.3, we provide the proof of Lemma 5.4.

## D.1  PROOF OF LEMMA 5.2

**Lemma D.1** (Formal version of Lemma 5.2). *For 2-layer ReLU activated neural network, suppose $m = \Omega(d \log(16n/\rho))$, then the following*

- $\|W_0\|_2 = O(\sqrt{m})$.

- $|f(W, x_i)| = O(1)$, *for $i \in [n]$.*

- $\|J_{W_0, x_i}\|_F = O(1)$, *for $i \in [n]$.*

*holds with prob. $\geq 1 - \rho/2$.*

*Proof.* (a) The first term can be seen in Corollary 5.35 of Vershynin (2010). Notice that $W_0 \in \mathbb{R}^{m \times d}$ is a Gaussian random matrix, the Corollary gives

$$\Pr[\|W_0\|_2 \leq \sqrt{m} + \sqrt{d} + t] \geq 1 - 2e^{-\frac{t^2}{2}}.$$

Let us set $m = \max\{d, \sqrt{2 \log(8/\rho)}\}$, it gives $\|W_0\|_2 \leq 3\sqrt{m}$ with probability $1 - \rho/4$.

(b) For the second term, first, $a_r, r \in [m]$ are Rademacher variables, thereby 1-sub-Gaussian, so with probability $1 - 2e^{-mt^2/2}$ we have $\frac{1}{m}|\sum_{r=1}^m a_r| \leq t$. This means if we take $m = \Omega(\log(16/\rho))$,

$$\Pr[\frac{1}{\sqrt{m}} \sum_{r=1}^m a_r = O(1)] \geq 1 - \frac{\rho}{8}. \tag{5}$$

Next, the vector $v_i = W_0^\top x_i \in \mathbb{R}^m$ is standard Gaussion vector. Write $a = \begin{bmatrix} a_1 & a_2 & \cdots & a_m \end{bmatrix}^\top$, since activation function $\phi_b$ is 1-Lipschitz, with a vector $a$ fixed, the function

$$\Phi : \mathbb{R}^m \to \mathbb{R}, v_i \mapsto \frac{1}{\sqrt{m}} a^\top \phi_b(v_i) = f(W_0, x_i)$$

has a Lipschitz parameter of $1/\sqrt{m}$.

Due to the concentration of a Lipschitz function under Gaussian variables (Theorem 2.26 in Wainwright (2019)),

$$\Pr[|\Phi(v_i) - \mathbb{E}_{W_0}(\Phi(v_i))| \geq t] \leq 2e^{-\frac{mt^2}{2}},$$

which means if $m = \Omega(\log(16n/\rho))$,

$$\left| \frac{1}{\sqrt{m}} a^\top \phi_b(W_0 x_i) - \frac{1}{\sqrt{m}} \left( \sum_{r \in [m]} a_r \right) \mathbb{E}_{w \sim \mathcal{N}(0, I_d)}[\phi_b(\langle w, x_i \rangle)] \right| = O(1) \tag{6}$$

---

**Algorithm 5** Our training algorithm, Formal version of Algorithm 1

---

1: **procedure** OURALGORITHM($\{x_i\}_{i \in [n]}, \epsilon$)
2:     /\*Initialization\*/
3:     Randomly pick $W(0)$
4:     TREE.INIT($\{(W_0)_r\}_{r \in [m]}, m, \{x_i\}_{i \in [n]}, n, d$)             $\triangleright$ Alg. 2
5:     $T \leftarrow \log(1/\epsilon), b \leftarrow \sqrt{0.48 \log m}$
6:     /\*Iterative Algorithm\*/
7:     **for** $t = 1 \to T$ **do**
8:         /\*Three computation tasks\*/
9:         /\*Step 1, Sketch computing\*/
10:        Implicitly write down the Jacobian matrix $J_t \in \mathbb{R}^{n \times md}$
11:        Let $A = J_t^\top$
12:        $\epsilon_{\text{sketch}} \leftarrow 0.1$
13:        $\delta_{\text{sketch}} \leftarrow 1/\operatorname{poly}(n)$
14:        $s_{\text{sketch}} \leftarrow n \operatorname{poly}(\epsilon_{\text{sketch}}^{-1}, \log(n/\delta_{\text{sketch}}))$
15:        Find sketching matrix $S \in \mathbb{R}^{s_{\text{sketch}} \times md}$ of $A$
16:        **for** $i = 1 \to n$ **do**
17:            $Q_i \leftarrow$ TREE.QUERY($i, b$)           $\triangleright Q_i \subset [m]$
18:                                               $\triangleright$ Theorem G.2 implies $|Q_i| = O(m^{0.76})$
19:            Let $D_i \in \mathbb{R}^{m \times m}$ denote a matrix where $(D_i)_{j,j} = 1$ if $j \in Q_i$
20:            Let $D_i \otimes I_d$ denote an $md \times md$ matrix
21:            $B_{*,i} \leftarrow S \cdot (D_i \otimes I_d) \cdot A_{*,i}$          $\triangleright S$ is a sketching matrix
22:        **end for**
23:        Let $Q = \cup_i Q_i$
24:        Let $D$ denote the diagonal version of $Q$
25:        /\*Step 2, Iterative regression\*/
26:        Compute $R \in \mathbb{R}^{n \times n}$ such that $SAR$ has orthonormal columns via QR decomposition
27:        $\tau \leftarrow 1$
28:        Compute $f_t$ based on $Q$
29:        Compute $y_{\text{reg}} \leftarrow f_t - y$
30:        $\epsilon_{\text{reg}} \leftarrow \frac{1}{6}\sqrt{\frac{\lambda}{n}}$
31:        **while** $\|A^\top(D \otimes I_d)ARz_t - y_{\text{reg}}\|_2 \geq \epsilon_{\text{reg}}$ **do**
32:            $z_{t+1} \leftarrow z_t - (R^\top A^\top(D \otimes I_d)AR)^\top (R^\top A^\top(D \otimes I_d)ARz_t - R^\top y_{\text{reg}})$
33:            $\tau \leftarrow \tau + 1$
34:        **end while**
35:        Compute $g_t \leftarrow z_t$
36:        /\*Step 3, Implicit weight maintenance\*/
37:        /\* $W_{t+1} \leftarrow W_t - J_t^\top g_t$ \*/
38:        Let $K \subset [m]$ denote the set of coordinates, we need to change the weights
39:                                          $\triangleright$ Theorem G.2 implies $|K| = O(m^{0.76}n)$
40:        **for** $r \in K$ **do**
41:            Compute $(W_{t+1})_r$           $\triangleright (W_{t+1})_r \in \mathbb{R}^d$
42:            TREE.UPDATE($(W_{t+1})_r, r$)          $\triangleright$ Alg. 3
43:        **end for**
44:     **end for**
45: **end procedure**

---

holds at the same time for all $i \in [n]$ with probability $1 - \frac{\rho}{8}$.

We know

$$|\mathbb{E}_{w \sim \mathcal{N}(0, I_d)}[\phi_b(wx_i)]| \leq |\phi_b(0)| + \mathbb{E}_{\xi \sim \mathcal{N}(0,1)}[|\xi|] = O(1). \tag{7}$$

Plugging in Eq. (5), (7) into Eq. (6), we see that once $m = \Omega(\log(16n/\rho))$, then with probability $1 - \rho/4$, for all $i \in [n]$,

$$|f(W_0, x_i)| = |\frac{1}{\sqrt{m}}a^\top \phi_b(W_0 x_i)| = O(1).$$

(c) Let $d_{W,x} = \phi_b'(Wx)$ denote the element-wise derivative of the activation function, since $\phi_b$ is 1-Lipschitz, we have $\|d_{W,x}\|_\infty = O(1)$. Note that $J_{W,x} = \frac{1}{\sqrt{m}}((d_{W,x} \circ a)x^\top)$ where $\circ$ denotes the element-wise product, we can easily know

$$\|J_{W,x_i}\|_F \leq \frac{1}{\sqrt{m}} \cdot \|\text{Diag}(d)\|_2 \cdot \|a\|_2 \cdot \|x\|_2 = O(1).$$

$\square$

## D.2 PROOF OF LEMMA 5.3

**Lemma D.2** (Shifted Perturbation Lemma, formal version of Lemma 5.3)**.** *For 2-layer ReLU activated neural network. Suppose the shifted parameter is $b$ ($b \geq 0$). Let $R_0 > 0$ be a parameter. Suppose*

$$m \geq \Omega(1) \cdot \max\{b^2 R_0^2, n^2 R_0^2 \lambda^{-2}, n\lambda^{-1} \log(n/\rho)\},$$

*then with prob. $\geq 1 - \rho - n^2 \cdot \exp\left(-m \cdot \min\{c'e^{-b^2/2}, \frac{R_0}{10\sqrt{m}}\}\right)$, for every $W \in \mathbb{R}^{d \times m}$ satisfying $\max_{r \in [m]} \|w_r - w_r(0)\|_2 \leq R_0/\sqrt{m}$, the following holds*

$$\|G_W - G_{W_0}\|_F \leq \lambda/2, \qquad \lambda_{\min}(G_W) \geq \lambda/2.$$

*Proof.* We use Lemma A.16 by setting $R = R_0/\sqrt{m}$ (that lemma require that $R \leq 1/b$) and letting $W = [\ w_1 \quad w_2 \quad \cdots \quad w_m\ ]$.

Since $R_0/\sqrt{m} \leq 1/b$, then we have $m \geq R_0^2 b^2$ (this is the corresponding to the first term of $m$ lower bound in lemma statement).

Note $H(W)$ is essentially $G_W$, and $\|w_r(t) - w_r(0)\|_2 \leq R$ for any $r$, thus by Lemma A.16, we have

- $\|G_W - G_0\|_F \leq n \cdot \min\{ce^{-b^2/2}, 3R\} = n \cdot \min\{ce^{-b^2/2}, 3R_0/\sqrt{m}\}$ with prob.

  $$1 - n^2 \exp(-m \cdot \min\{c'e^{-b^2/2}, R/10\}) = 1 - n^2 \exp(-m \cdot \min\{c'e^{-b^2/2}, \frac{R_0}{10\sqrt{m}}\}),$$

- $\lambda_{\min}(G_W) \geq \frac{3}{4}\lambda - n\min\{ce^{-b^2/2}, 3R\} = \frac{3}{4}\lambda - n\min\{ce^{-b^2/2}, 3R_0/\sqrt{m}\}$ with prob.

  $$1 - \rho - n^2 \exp(-m \cdot \min\{c'e^{-b^2/2}, R/10\}) = 1 - \rho - n^2 \exp(-m \cdot \min\{c'e^{-b^2/2}, \frac{R_0}{10\sqrt{m}}\}).$$

Then it remains to prove

$$n \cdot \min\{ce^{-b^2/2}, 3R_0/\sqrt{m}\} \leq \frac{\lambda}{2}.$$

Since $m \geq \Omega(n^2 R_0^2 \lambda^{-2})$, we have $3nR_0/\sqrt{m} \leq \frac{\lambda}{2}$, which finishes the proof. $\square$

## D.3 PROOF OF LEMMA 5.4

**Lemma D.3** (The shifted NTK version of Lemma C.4 in Brand et al. (2021), formal version of Lemma 5.4)**.** *Suppose $R_0 \geq 1$ and $m = \widetilde{\Omega}(n^2 R_0^2)$. Then for every $w \in \mathbb{R}^{d \times m}$ satisfying $\max_{r \in [m]} \|w_r - w_r(0)\|_2 \leq R_0/\sqrt{m}$, the following holds*

- $\|W - W_0\| = O(R_0)$,

- $\|J_{W,x_i} - J_{W_0,x_i}\|_2 = \widetilde{O}(R_0^{1/2}/m^{1/4})$ and $\|J_W - J_{W_0}\|_F = \widetilde{O}(n^{1/2}R_0^{1/2}/m^{1/4})$,

- $\|J_W\|_F = O(\sqrt{n})$,

*with prob. $\geq 1 - \rho$. The randomness comes from the initialization of $W_0$.*

*Proof.* (1) The first claim follows from

$$\|W - W_0\| \leq \|W - W_0\|_F$$

$$= \Big( \sum_{r=1}^{m} \|w_r - w_r(0)\|_2^2 \Big)^{1/2}$$

$$\leq \sqrt{m} \cdot R_0/\sqrt{m}$$

$$= R_0.$$

where the first step comes from $\| \cdot \| \leq \| \cdot \|_F$, the second step comes from definition of Frobenius norm, the third step comes from $\|w_r - w_r(0)\|_2 \leq R_0/\sqrt{m}$, and the last step comes from canceling $\sqrt{m}$.

(2) For the second claim, we have for any $i \in [n]$

$$\|J_{W,x_i} - J_{W_0,x_i}\|^2 = \frac{1}{m} \sum_{r=1}^{m} a_r^2 \cdot \|x_r\|_2^2 \cdot |\mathbf{1}_{\langle w_r, x_i \rangle \geq b} - \mathbf{1}_{\langle w_r(0), x_i \rangle \geq b}|^2$$

$$= \frac{1}{m} \sum_{r=1}^{m} |\mathbf{1}_{\langle w_r, x_i \rangle \geq b} - \mathbf{1}_{\langle w_r(0), x_i \rangle \geq b}|. \tag{8}$$

The second equality follows from $a_r \in \{-1, 1\}$, $\|x_i\|_2 = 1$ and

$$s_{i,r} := |\mathbf{1}_{\langle w_r, x_i \rangle \geq b} - \mathbf{1}_{\langle w_r(0), x_i \rangle \geq b}| \in \{0, 1\}. \tag{9}$$

We define the event $A_{i,r}$ as

$$A_{i,r} = \big\{ \exists \widetilde{w} \ : \ \|\widetilde{w} - w_r(0)\| \leq R_0/\sqrt{m}, \quad \mathbf{1}_{\langle \widetilde{w}, x_i \rangle \geq b} \neq \mathbf{1}_{\langle w_r(0), x_i \rangle \geq b} \big\}.$$

It is not hard to see $A_{i,r}$ holds if and only if $\langle w_r(0), x_i \rangle \in [b - R_0/\sqrt{m}, b + R_0/\sqrt{m}]$. Since $w_r(0)$ is sampled from Gaussian $\mathcal{N}(0, I_d)$ and $\|x_i\| = 1$, we have $\langle w_r(0), x_i \rangle$ is sampled from Gaussian $\mathcal{N}(0, 1)$, thus by the anti-concentration of Gaussian (see Lemma A.15), we have

$$\mathbb{E}[s_{i,r}] = \Pr[A_{i,r}] = \Pr[\, \langle w_r(0), x_i \rangle \in [b - R_0/\sqrt{m}, b + R_0/\sqrt{m}]\, ]$$

$$\leq \Pr[\, \langle w_r(0), x_i \rangle \in [-R_0/\sqrt{m}, R_0/\sqrt{m}]\, ]$$

$$\leq \frac{4}{5} R_0/\sqrt{m}.$$

Thus we have

$$\Pr\left[ \sum_{i=1}^{m} s_{i,r} \geq (t + 4/5) R_0 \sqrt{m} \right] \leq \Pr\left[ \sum_{i=1}^{m} (s_{i,r} - \mathbb{E}[s_{i,r}]) \geq t R_0 \sqrt{m} \right]$$

$$\leq 2 \exp\left( -\frac{2t^2 R_0^2 m}{m} \right)$$

$$= 2 \exp(-t^2 R_0^2)$$

$$\leq 2 \exp(-t^2). \tag{10}$$

holds for any $t > 0$. The second inequality is due to the Hoeffding bound (see Lemma A.14), the last inequality is because $R_0 > 1$. Taking $t = 2\log(n/\rho)$ and using union bound over $i$, with prob. $\geq 1 - \rho$,

$$\|J_{W,x_i} - J_{W_0,x_i}\|_2^2 = \frac{1}{m} \sum_{r=1}^{m} s_{i,r} \leq \frac{1}{m} \cdot 2 \log(n/\rho) R_0 \sqrt{m} = \widetilde{O}(R_0/\sqrt{m})$$

holds for all $i \in [n]$. The first equality comes from Eq. (8) and Eq. (9), the second inequality comes from Eq. (10). Thus we conclude with

$$\|J_{W,x_i} - J_{W_0,x_i}\|_2 = \widetilde{O}(R_0^{1/2}/m^{1/4}) \quad \text{and} \quad \|J_W - J_{W_0}\|_F = \widetilde{O}(n^{1/2} R_0^{1/2}/m^{1/4}). \tag{11}$$

(3) The thrid claim follows from

$$\|J_W\|_F \leq \|J_{W_0}\|_F + \|J_W - J_{W_0}\|_F$$

$$\leq O(\sqrt{n}) + \|J_W - J_{W_0}\|_F$$
$$\leq O(\sqrt{n}) + \widetilde{O}(n^{1/2} R_0^{1/2}/m^{1/4})$$
$$= O(\sqrt{n}).$$

where the 1st step is due to triangle inequality, the 2nd step is due to the third claim in Lemma 5.2, the 3rd step is due to Eq. (11), and the last step is due to $m = \widetilde{\Omega}(R_0^2 n^2)$.

$\square$

# E  INDUCTION

Section 5 has defined the induction hypothesis (see Definition 5.5) and given a lemma (see Lemma 5.6) to prove that induction hypothesis holds for all time with high probability, but left its proof to this section. Here, we present and prove the following Lemma E.1, the formal version of Lemma 5.6, and then the crucial Theorem 5.1 holds straightforwardly. We divided the proof of each part of the lemma in Section E.1 and Section E.2, and combine them in Section E.3.

**Lemma E.1** (Formal version of Lemma 5.6). *Define $R_0 \approx n/\lambda$. With probability at least $1 - \frac{5}{2}\rho - n^2 \cdot \exp\left(-m \cdot \min\{c'e^{-b^2/2}, \frac{R_0}{10\sqrt{m}}\}\right)$ of the initial weights $W_0$, for every $t > 0$, if*

- $\|f_t - y\|_2 \leq \frac{1}{2}\|f_{t-1} - y\|_2$
- $\max_{r \in [m]} \|w_r(t) - w_r(0)\|_2 \leq R_0/\sqrt{m}$

*then*

- $\|f_{t+1} - y\|_2 \leq \frac{1}{2}\|f_t - y\|_2$
- $\max_{r \in [m]} \|w_r(t+1) - w_r(0)\|_2 \leq R_0/\sqrt{m}$

*also holds.*

## E.1  PROOF OF LEMMA E.1: THE FIRST LEMMA

As stated in the previous subsection, we use induction. Here we need to break the induction step (Lemma E.1) into two separate steps, Lemma E.2 and Lemma E.3. Each separated induction step corresponds to prove one part in the Lemma E.1. We first prove the first part of Lemma E.1.

**Lemma E.2** (Part 1 of Lemma E.1). *Suppose initial weights $W_0$ satisfies the restriction of Lemma 5.2, 5.3 and 5.4, then for any fixed $t$, if*

- $\|f_t - y\|_2 \leq \frac{1}{2}\|f_{t-1} - y\|_2$ *holds*
- $\max_{r \in [m]} \|w_r(t) - w_r(0)\|_2 \leq R_0/\sqrt{m}$ *holds*

*Then we have*

- $\|f_{t+1} - y\|_2 \leq \frac{1}{2}\|f_t - y\|_2$ *holds.*

This proof is similar to Brand et al. (2021), for the completeness, we still provide the details here.

*Proof.* We prove the first claim holds for time $t + 1$. Define

$$J_{t,t+1} = \int_0^1 J\Big((1-s)W_t + sW_{t+1}\Big)\mathrm{d}s,$$

and denote $g^\star = (J_t J_t^\top)^{-1}(f_t - y)$ to be the optimal solution to Eq. (4), then we have

$$\|f_{t+1} - y\|_2$$
$$= \|f_t - y + (f_{t+1} - f_t)\|_2$$

$$= \|f_t - y + J_{t,t+1}(W_{t+1} - W_t)\|_2$$

$$= \|f_t - y - J_{t,t+1} J_t^\top g_t\|_2$$

$$= \|f_t - y - J_t J_t^\top g_t + J_t J_t^\top g_t - J_{t,t+1} J_t^\top g_t\|_2$$

$$\le \|f_t - y - J_t J_t^\top g_t\|_2 + \|(J_t - J_{t,t+1}) J_t^\top g_t\|_2$$

$$\le \|f_t - y - J_t J_t^\top g_t\|_2 + \|(J_t - J_{t,t+1}) J_t^\top g^\star\|_2 + \|(J_t - J_{t,t+1}) J_t^\top (g_t - g^\star)\|_2, \tag{12}$$

where the 2nd step is from the definiton of $J_{t,t+1}$ and simple calculus, the 3rd step is from the updating rule of the algorithm, the 5th step is due to triangle inequality, and the sixth step is because triangle inequality.

For the first quantity in Eq. (12), we have

$$\|J_t J_t^\top g_t - (f_t - y)\|_2 \le \frac{1}{6} \|f_t - y\|_2, \tag{13}$$

since $g_t$ is an $\epsilon_0 (\epsilon_0 \le \frac{1}{6})$ approximate solution to regression problem (4).

For the second quantity in Eq. (4), we have

$$\|(J_t - J_{t,t+1}) J_t^\top g^\star\|_2 \le \|(J_t - J_{t,t+1})\| \cdot \|J_t^\top g^\star\|_2$$

$$= \|(J_t - J_{t,t+1})\| \cdot \|J_t^\top (J_t J_t^\top)^{-1} (f_t - y)\|_2$$

$$\le \|(J_t - J_{t,t+1})\| \cdot \|J_t^\top (J_t J_t^\top)^{-1}\| \cdot \|(f_t - y)\|_2 \tag{14}$$

where the 1st step is due to matrix spectral norm, the 2nd step is because the definition of $g^*$, and the 3rd step relies on matrix spectral norm.

We bound these term separately. First,

$$\|J_t - J_{t,t+1}\| \le \int_0^1 \|J((1-s)W_t + sW_{t+1}) - J(W_t)\| \mathrm{d}s$$

$$\le \int_0^1 \left( \|J((1-s)W_t + sW_{t+1}) - J(W_0)\| + \|J(W_0) - J(W_t)\| \right) \mathrm{d}s$$

$$\le \widetilde{O}(R_0^{1/2} n^{1/2} / m^{1/4}), \tag{15}$$

where the 1st step comes from simple calculus, the 2nd step comes from triangle inequality, and the 3rd step comes from the second claim in Lemma 5.4 and the fact that

$$\|(1-s)w_r(t) + sw_r(t+1) - w_0\|_2 \le (1-s)\|w_r(t) - w_r(0)\|_2 + s\|w_r(t+1) - w_r(0)\|_2$$

$$\le R_0/\sqrt{m}.$$

Then, we have

$$\|J_t^\top (J_t J_t^\top)^{-1}\| = \frac{1}{\sigma_{\min}(J_t^\top)} \le \sqrt{2/\lambda} \tag{16}$$

where the 2nd step comes from $\sigma_{\min}(J_t) = \sqrt{\lambda_{\min}(J_t^\top J_t)} \ge \sqrt{\lambda/2}$ (see Lemma 5.3).

Combining Eq. (14), (15) and (16), we have

$$\|(J_t - J_{t,t+1}) J_t^\top g^\star\|_2 \le \widetilde{O}(R_0^{1/2} \lambda^{-1/2} n^{1/2} / m^{1/4}) \|f_t - y\|_2$$

$$= \widetilde{O}(\lambda^{-1} n m^{-1/4}) \|f_t - y\|_2$$

$$\le \|f_t - y\|/6, \tag{17}$$

since $m = \widetilde{\Omega}(\lambda^{-4} n^4)$.

Let us consider the third term in Eq. (12),

$$\|(J_t - J_{t,t+1}) J_t^\top (g_t - g^\star)\|_2 \le \|J_t - J_{t,t+1}\| \cdot \|J_t^\top\| \cdot \|g_t - g^\star\|_2 \tag{18}$$

by matrix norm. Moreover, one has

$$\frac{\lambda}{2} \|g_t - g^\star\|_2 \le \lambda_{\min}(J_t J_t^\top) \|g_t - g^\star\|_2$$

$$\leq \|J_t J_t^\top g_t - J_t J_t^\top g^\star\|_2$$
$$= \|J_t J_t^\top g_t - (f_t - y)\|_2$$
$$\leq \sqrt{\lambda/n} \cdot \|f_t - y\|_2, \tag{19}$$

where 1st step comes from $\lambda_{\min}(J_t J_t^\top) = \lambda_{\min}(G_t) \geq \lambda/2$ (see Lemma 5.3), the 2nd step is because simple linear algebra, the 3rd step is because the definition of $g^*$, and the last step is because $g_t$ is an $\epsilon_0$-approximate solution to $\min_{g_t} \|J_t J_t^\top g_t - (f_t - y)\|$ and $\epsilon_0 \leq \sqrt{\lambda/n}$.

Consequently, we have

$$\|(J_t - J_{t,t+1})J_t^\top (g_t - g^\star)\|_2 \leq \|J_t - J_{t,t+1}\| \cdot \|J_t^\top\| \cdot \|g_t - g^\star\|_2$$
$$\leq \widetilde{O}(R_0^{1/2} n^{1/2} m^{-1/4}) \cdot \sqrt{n} \cdot \frac{2}{\sqrt{n\lambda}} \cdot \|f_t - y\|_2$$
$$= \widetilde{O}(n\lambda^{-1} m^{-1/4}) \cdot \|f_t - y\|_2$$
$$\leq \frac{1}{6} \|f_t - y\|_2, \tag{20}$$

where the 1st step is because of matrix spectral norm, the 2nd step comes from Eq. (15), (19) and the fact that $\|J_t\| \leq O(\sqrt{n})$ (see Lemma 5.4), and the last step comes from the $m = \Omega(n^4\lambda^{-4})$. Combining Eq. (12), (13), (17), and (20), we have proved the first claim, i.e.,

$$\|f_{t+1} - y\|_2 \leq \frac{1}{2}\|f_t - y\|_2. \tag{21}$$

Thus, we complete the proof. $\qquad\square$

### E.2 PROOF OF LEMMA E.1: THE SECOND LEMMA

We now move to the second part for Lemma E.1. We show it in Lemma E.3.

**Lemma E.3** (Part 2 of Lemma E.1). *Suppose initial weights $W_0$ satisfies the restriction of Lemma 5.2, 5.3 and 5.4, then for any fixed $t$, if*

- $\|f_t - y\|_2 \leq \frac{1}{2}\|f_{t-1} - y\|_2$ *holds*

- $\max_{r \in [m]} \|w_r(t) - w_r(0)\|_2 \leq R_0/\sqrt{m}$ *holds*

*Then we have*

- $\max_{r \in [m]} \|w_r(t+1) - w_r(0)\|_2 \leq R_0/\sqrt{m}$ *holds*

This proof is similar to Brand et al. (2021), for the completeness, we still provide the details here.

*Proof.* First, we have

$$\|g_t\|_2 \leq \|g^\star\|_2 + \|g_t - g^\star\|_2$$
$$\leq \|(J_t J_t^\top)^{-1}(f_t - y)\|_2 + \|g_t - g^\star\|_2$$
$$\leq \|(J_t J_t^\top)^{-1}\| \cdot \|(f_t - y)\|_2 + \|g_t - g^\star\|_2$$
$$\leq \frac{2}{\lambda} \cdot \|f_t - y\|_2 + \frac{2}{\sqrt{n\lambda}} \cdot \|f_t - y\|_2$$
$$\lesssim \frac{1}{\lambda} \cdot \|f_t - y\|_2, \tag{22}$$

where the 1st step relies on triangle inequality, the 2nd step replies on the definition of $g^*$, the 3rd step uses matrix norm, the 4th step comes from Eq. (19) and the last step uses the obvious fact that $1/\sqrt{n\lambda} \leq 1/\lambda$.

Hence, for any $0 \leq k \leq t$ and $r \in [m]$, if we use $g_{k,i}$ to denote the $i^{th}$ indice of $g_k$, then we have

$$\|w_r(k+1) - w_r(k)\|_2 = \|(J_k^\top g_k)_r\|_2$$

$$= \left\| \sum_{i=1}^{n} \frac{1}{\sqrt{m}} a_r x_i^\top \mathbf{1}_{\langle w_r(t), x_i \rangle \geq b} g_{k,i} \right\|_2$$

$$\leq \frac{1}{\sqrt{m}} \sum_{i=1}^{n} |g_{k,i}|$$

$$\leq \frac{\sqrt{n}}{\sqrt{m}} \|g_k\|_2$$

$$\lesssim \frac{\sqrt{n}}{\sqrt{m}} \cdot \frac{1}{2^k \lambda} \|f_0 - y\|_2$$

$$\lesssim \frac{n}{\sqrt{m}\lambda} \cdot \frac{1}{2^k}, \tag{23}$$

where the 1st step is because of the updating rule, the 2nd step is because of the definition of $J_k$, the 3rd step is because of triangle inequalities and the fact that $a_r = \pm 1$, $\|x_r\|_2 = 1$, the 4th step comes is because of Cauchy-Schwartz inequality, the 5th step is because of Eq. (21) and Eq. (22), and the last step is because of the fact that $\|f_0 - y\|_2 \leq O(\sqrt{n})$ (see Lemma 5.2, $f_0(x_i) = O(1)$ for any $i \in [n]$, thus $\|f_0 - y\|_2 = \sqrt{\sum_{i=1}^{n}(f(x_i) - y_i)^2} = O(\sqrt{n})$). Consequently, we have

$$\|w_r(t+1) - w_r(0)\|_2 \leq \sum_{k=0}^{t} \|w_r(k+1) - w_r(k)\|_2 \lesssim \sum_{k=0}^{t} \frac{n}{\sqrt{m}\lambda} \cdot \frac{1}{2^k} \lesssim \frac{R_0}{\sqrt{m}},$$

where the 1st step is because of triangle inequality, the 2nd step is because of Eq. (23), and the last step is because of simple summation.

Thus we also finish the proof of the second claim. $\qquad\square$

### E.3    Proof of Lemma E.1: combination

We use Lemma E.2 and Lemma E.3 to prove Lemma E.1.

*Proof.* Since the probability of initial weight $W_0$ satisfies the restriction of Lemma 5.2, Lemma 5.3 and Lemma 5.4 is $1 - \rho/2$, $1 - \rho - n^2 \cdot \exp\left(-m \cdot \min\{c'e^{-b^2/2}, \frac{R_0}{10\sqrt{m}}\}\right)$, $1 - \rho$ respectively, by union bound, the probability of they all happen is at least

$$1 - \frac{5}{2}\rho - n^2 \cdot \exp\left(-m \cdot \min\{c'e^{-b^2/2}, \frac{R_0}{10\sqrt{m}}\}\right)$$

In this case, for any fixed $t$, combining Lemma E.2 and Lemma E.3, if

- $\|f_t - y\|_2 \leq \frac{1}{2}\|f_{t-1} - y\|_2$ holds,
- $\max_{r \in [m]} \|w_r(t) - w_r(0)\|_2 \leq R_0/\sqrt{m}$ holds

then we have

- $\|f_{t+1} - y\|_2 \leq \frac{1}{2}\|f_t - y\|_2$ holds.
- $\max_{r \in [m]} \|w_r(t+1) - w_r(0)\|_2 \leq R_0/\sqrt{m}$ holds

Thus by induction, with prob. $\geq 1 - \frac{5}{2}\rho - n^2 \cdot \exp\left(-m \cdot \min\{c'e^{-b^2/2}, \frac{R_0}{10\sqrt{m}}\}\right)$,

$$\|f_t - y\|_2 \leq \frac{1}{2}\|f_{t-1} - y\|_2$$

holds for all $t$, hence finished the proof of Lemma E.1. $\qquad\square$

### E.4 NUMBER OF ITERATIONS FOR ITERATIVE REGRESSION

**Lemma E.4.** *The iterative regression in our fast training algorithm requires $O(\log(n/\lambda))$ iterations.*

*Proof.* By Lemma D.1, $\|J_t J_t^\top\| = \|G_t\| = O(n)$ and $\lambda_{\min}(J_t J_t^\top) = \lambda_{\min}(G_t) \geq O(\lambda)$. Let $\epsilon_{\mathrm{reg}}$ be chosen as Algorithm 5.

Thus by Corollary A.19, the number of iterations needed by the iterative regression is

$$O(\log(\kappa(J_t^\top)/\epsilon_{\mathrm{reg}})) = O(\log(\sqrt{n/\lambda}/\sqrt{\lambda/n}))$$
$$= O(\log(n/\lambda)).$$

$\square$

## F MORE RUNNING TIME DETAILS

Section 6 analyzes the running time of our algorithm. It shows that when $m$ is large enough, the running time of CPI is $o(mnd) + \widetilde{O}(n^3)$, and with FMM, the CPI can be reduced to $o(mnd) + \widetilde{O}(n^\omega)$.

In this section, we give the specific time complexity hidden by $o(mnd)$, and also give the complete algorithm representation of our training algorithm. It will show that when $m$ is large enough, the CPI is $\widetilde{O}(m^{1-\alpha}nd)$. Similar with Section 6, we first present Theorem F.1, the running time result. We then provide three lemmas (Lemma F.2, Lemma F.3 and Lemma F.4) to prove our main theorem. Our main running time result is the following:

**Theorem F.1** (Running time part of Theorem 1.1, formal version of Theorem 6.1)**.** *The CPI is $\widetilde{O}(m^{1-\alpha}nd + n^3)$, and the running time for shrinking the training loss to $\epsilon$ is $\widetilde{O}((m^{1-\alpha}nd + n^3)\log(1/\epsilon))$.*

*Using FMM, the CPI is $\widetilde{O}(m^{1-\alpha}nd + n^\omega)$, the running time is $\widetilde{O}((m^{1-\alpha}nd + n^\omega)\log(1/\epsilon))$. Note that $\omega$ is the exponent of matrix multiplication. Currently, $\omega \approx 2.373$.*

*Proof.* Combining Lemma F.2, Lemma F.3 and Lemma F.4, the computation time of each iteration is

$$\widetilde{O}(n^2 m^{0.76} d) + \widetilde{O}(n m^{0.76} d + n^3) + O(n^2 m^{0.76}(d + \log m))$$
$$= \widetilde{O}(n^2 m^{0.76} d + n^3 + n^2 m^{0.76} d)$$
$$= \widetilde{O}(n^2 m^{0.76} d + n^3),$$

where the first step comes from hiding $\log m$ on $\widetilde{O}$, the second step comes from simple merging. And if using FMM, similarly the running time is $\widetilde{O}(n^2 m^{0.76} d + n^\omega)$.

By Theorem 5.1, we have: The time to reduce the training loss to $\epsilon$ is $\widetilde{O}((n^2 m^{0.76} d + n^3)\log(1/\epsilon))$. Taking advantage of FMM, the time is $\widetilde{O}((n^2 m^{0.76} d + n^\omega)\log(1/\epsilon))$.

Further, for example, if $m = n^c$ where $c$ is some large constant, then $n^2 m^{0.76} d \leq n m^{1-\alpha} d$ where $\alpha \in [0.1, 0.24)$. Hence the time of each iteration is $\widetilde{O}(m^{1-\alpha}nd + n^3)$, and the time to reduce the training loss to $\epsilon$ is $\widetilde{O}((m^{1-\alpha}nd + n^3)\log(1/\epsilon))$. Taking advantage of FMM, the time is $\widetilde{O}((m^{1-\alpha}nd + n^\omega)\log(1/\epsilon))$. Thus we complete the proof. $\square$

For the rest of this section, we provide detailed analysis for the steps. In Section F.1 we analyse the sketch computing step. In Section F.2 we analyse the iterative regression step. In Section F.3 we analyse the implicit weight maintenance step.

### F.1 SKETCH COMPUTING

We delicate to prove the lemma that formally analyzes the running time of the sketch computing process in Algorithm 5 to show its time complexity.

**Lemma F.2** (Sketch computing, formal version of Lemma 6.3). *The sketch computing process of Algorithm 5 (from line 10 to line 23) runs in time $\widetilde{O}(m^{0.76}n^2d)$.*

*Proof.* In the sketch computing process, by Corollary A.22, only $O(m^{0.76}d)$ entries of each column of $A$ is nonzero, thus calculating each column of $B$ takes $O(m^{0.76}dt)$ time, where $t$ is the number of rows of $B$. And according to Lemma A.11,

$$t = n \operatorname{poly}(\log(n/\delta_{\mathrm{sketch}}))/\epsilon_{\mathrm{sketch}}^2$$
$$= O(n \operatorname{poly}(\log(n/\delta_{\mathrm{sketch}}))).$$

Since

$$\epsilon_{\mathrm{sketch}} = 0.1 \quad \text{and} \quad \delta_{\mathrm{sketch}} = \frac{1}{\operatorname{poly}(n)},$$

the whole for-loop runs in time $O(n^2 m^{0.76}d \operatorname{poly}(\log(n)))$. $\qquad\square$

### F.2 ITERATIVE REGRESSION

We delicate to prove a lemma that formally analyzes the running time of the iterative regression process in Algorithm 5 to show its time complexity.

**Lemma F.3** (Iterative regression, formal version of Lemma 6.4). *The iterative regression of Algorithm 5 (from line 26 to line 35) runs in time*

$$\widetilde{O}(nm^{0.76}d + n^3).$$

*Taking advantage of* FMM, *the running time is*

$$\widetilde{O}(nm^{0.76}d + n^\omega),$$

*where $\omega$ is the exponent of matrix multiplication. Currently $\omega \approx 2.3713$ Alman et al. (2024a).*

*Proof.* The algorithm calculate $R$ using $QR$ decomposition in line 26 (Algorithm 5). This step will take $O(n^3)$ time. Taking advantage of FMM, it will take $O(n^\omega)$ time Alman et al. (2024a).

For the while-loop from line 31 (Algorithm 5), define $p$ as the number of iterations of the while-loop from line 31 (Algorithm 5), then

$$p = O(\log(n/\lambda))$$
$$= O(\log(\frac{n}{(\exp(-b^2/2) \cdot \frac{\delta}{100n^2})}))$$
$$= O(\log(n/\delta) + b^2)$$
$$= O(\log(n/\delta) + \log m)$$
$$= O(\log(mn/\delta)),$$

where the 1st step comes from Lemma E.4, the 2nd step comes from Theorem A.9, the 3rd step comes from identical transformation, and the 4th step comes from $b = \Theta(\sqrt{\log m})$.

And in each iteration, note that $R$ is $n \times n$, $A$ is $md \times n$, $S$ is $t \times md$,

- we have calculating $v = R^\top A^\top (D \otimes I_d)ARz_t - R^\top y_{\mathrm{reg}}$ takes
$$O(n^2 + m^{0.76}dn + m^{0.76}dn + n^2 + n^2) = O(m^{0.76}dn + n^2)$$
  time,

- and calculating $(R^\top A^\top (D \otimes I_d)AR)^\top v$ takes
$$O(n^2 + m^{0.76}dn + m^{0.76}dn + n^2) = O(m^{0.76}dn + n^2)$$
  time.

Thus each iteration in the while-loop from line 31 (Algorithm 5) takes $O(m^{0.76}dn + n^2)$ time, the total process of the iterative regression takes $O((m^{0.76}dn + n^2)\log(mn/\delta) + n^3)$ time.

Using FMM, the running time is $O((m^{0.76}dn + n^2)\log(mn/\delta) + n^\omega)$.

In our regime, $O(\log(n/\delta)) = O(\log m)$ since $m = \operatorname{poly}(n/\delta)$. Thus, we can hide the log factors in $\widetilde{O}$. $\qquad\square$

### F.3 IMPLICIT WEIGHT MAINTENANCE

We give a lemma that formally analyzes the running time of the implicit weight maintenance process in Algorithm 5 to show its time complexity.

**Lemma F.4** (Implicit weight maintenance, formal version of Lemma 6.5). *The implicit weight maintenance of Algorithm 5 (from line 38 to line 43) runs in time $O(n^2 m^{0.76}(d + \log m))$.*

*Proof.* Let us consider every iteration of the for loop starting at line 40 (Algorithm 5), since $(J_t)_r$ is $d \times n$, computing $W_{t+1}$ takes $O(nd)$ time. And by Lemma B.3, updating $W_{t+1}$ takes $O(n(d+\log m))$ time, thus each iteration takes $O(n(d + \log m))$ time. By Theorem G.2, $|K| = O(nm^{0.76})$, thus the whole implicit weight maintenance takes $O(n^2 m^{0.76}(d + \log m))$ time. □

## G COMBINATION

Theorem 1.1 shows that as long as the 2-layer neural network is broad enough, then there exists a training algorithm with sublinear running time and large converge probability. Theorem 5.1 gives an analysis about how large $m$ should be, but its result is based on $\lambda$, the minimal eigenvalue of $K$ [2], which is not straightforward.

In this section, we convert the bound of Theorem 5.1 into a bound only related to batch number $n$, data separability $\delta$ and tolerable probability of failure $\rho$.

**Definition G.1** (Two sparsity definitions). *We define sparsity of the 2-layer neural network to the number of activated neurons.*

*We define sparsity of a Jacobi matrix of 2-layer neural network as the maximal number of non-zero entries of a row in the Jacobi matrix $J$ ($J \in \mathbb{R}^{n \times md}$) of weights.*

We present the following theorem.

**Theorem G.2.** *For a 2-layer ReLU activated neural network. Suppose $m$ is the number of neurons, $d$ is the dimension of points, $n$ is represent the number of points, $\rho \in (0, 1/10)$ is the failure probability, and $\delta$ is the separability of data points.*

*For any real number $\overline{\alpha} \in (0, 1]$, let $b = \sqrt{0.5(1 - \overline{\alpha}) \log m}$, if*

$$m = \Omega((\delta^{-4} n^{12} \log^4(n/\rho))^{1/\overline{\alpha}})$$

*then the training algorithm in Algorithm 5 converges with prob. $\geq 1 - \frac{5}{2}\rho - n^2 \cdot \exp(-m \cdot \min\{c'e^{-b^2/2}, \frac{R}{10\sqrt{m}}\})$, and the sparsity of the neural network is*

$$O(m^{\frac{3+\overline{\alpha}}{4}})$$

*with probability $1 - n \cdot \exp(-\Omega(m \cdot \exp(-b^2/2)))$. Especially, for any given parameter $\epsilon_0 \in (0, 1/4]$, if we choose $\overline{\alpha} = 0.04$, the sparsity is $O(m^{0.76})$.*

*Proof.* From Theorem A.9, we know

$$\lambda \geq \exp(-b^2/2) \cdot \frac{\delta}{100n^2}.$$

Since by Theorem 5.1, we need

$$m = \Omega(\lambda^{-4} n^4 b^2 \log^2(n/\rho))$$

to make our algorithm converges, we need to choose

$$m = \Omega((\exp(b^2/2) \cdot 100n^2 \cdot \delta^{-1})^4 \cdot n^4 b^2 \log^2(n/\rho))$$
$$= \Omega(\exp(4 \cdot b^2/2) \cdot \delta^{-4} \cdot n^{12} b^2 \log^2(n/\rho))$$
$$= \Omega(m^{1-\overline{\alpha}} \cdot \delta^{-4} \cdot n^{12} \cdot (\log m) \cdot \log^2(n/\rho))$$

---

[2] See Section (3) for the definition of $K$.

where the final step is because $b = \sqrt{0.5(1 - \overline{\alpha}) \log m}$.

Suppose the constant hidden by $\Omega$ is $C$, then the above equation is equivalent to

$$m^{\overline{\alpha}} \geq C \cdot \delta^{-4} \cdot n^{12} \cdot (\log m) \cdot \log^2(n/\rho),$$

and since $m = \text{poly}(n)$, $\log m \leq \log^2 n$, thus as long as

$$m \geq (C\delta^{-4} n^{12} \log^4(n/\rho))^{1/\overline{\alpha}},$$

we have $m = \Omega(\lambda^{-4} n^4 b^2 \log^2(n/\rho))$, then by Theorem 5.1, our algorithm converges.

Then, according to Lemma A.21, the sparsity of this neural network is equal to

$$= O(m \cdot \exp(-b^2/2))$$
$$= m \cdot m^{-(1-\overline{\alpha})/4}$$
$$= m^{\frac{3+\overline{\alpha}}{4}}$$

where the second step is because $b = \sqrt{0.5(1 - \overline{\alpha}) \log m}$, for any $\overline{\alpha} \in (0, 1]$. $\qquad \square$

