# OpenReview forum: "Training Overparametrized Neural Networks in Sublinear Time"
_ICLR.cc/2025/Conference — Submitted to ICLR 2025_

### Official Review · Reviewer_HCtX · 2024-10-22

**Soundness:** 2
**Presentation:** 1
**Contribution:** 2
**Rating:** 3
**Confidence:** 3

**Summary:**

This paper proposes a second-order optimization algorithm based on the Gauss-Newton method to efficiently train an overparameterized 2-layer shifted ReLU network with a fixed second layer, using squared loss in the NTK regime. The challenge with second-order methods is their high cost-per-iteration (CPI), despite requiring fewer iterations for convergence compared to first-order methods. The authors aim to reduce CPI while maintaining fast convergence. Under the overparameterized setting, where the width $m$ is polynomial in the number of training points $n$ ($m = \mathrm{poly}(n)$) with input dimension $d$, the proposed method reduces CPI to $O(m^{1-\alpha}nd + n^3)$ for some constant $\alpha \in (0.01, 1)$, improving upon existing second-order methods with a CPI of $O(mnd + n^3)$. The method also achieves $O(\log(1/\epsilon))$ iterations to reach a training error below $\epsilon$, matching the convergence rate of existing second-order methods.

The key idea behind the algorithm is using a tree-based data structure to efficiently identify activated neurons, leveraging the sparsity of shifted ReLU activations. Combined with sketching techniques for solving the online regression problem from the Gauss-Newton method, this leads to a sublinear CPI.

**Strengths:**

The proposed algorithm achieves a CPI that is sublinear in the network width, which is an improvement over existing second-order methods in overparameterized settings. It is great that the paper provides formal proof of both the sublinear CPI and the ultra-fast convergence rate typical of second-order methods.

**Weaknesses:**

This paper has several critical weaknesses, including issues with the novelty of the tree-based data structure, poor writing, and a lack of practical implications.

1. **The proposed dynamic tree data structure lacks novelty and proper attribution to prior work.**
-  The dynamic tree data structure is not novel and lacks proper citation of prior work, such as [Alman et al., 2023]. The correlation tree proposed in [Alman et al., 2023] (Algorithms 1 and 2 therein) is nearly identical to the threshold search data structure used in this paper (Algorithms 2, 3, and 4). Notations and text are similar, but the authors do not properly cite this relevant work while claiming novelty.

2. **The writing is unpolished and the presentation of key concepts is imprecise.**
- The paper does not sufficiently describe the model setup or key notations in the main text, making it hard for readers to follow. For example, fundamental elements like the loss function and variables such as $J_t$, $f_t$, and $y$ (which appear on Line 247) are only defined in the appendix. Such definitions should be in the main text or referenced before their use to improve clarity.
- In Theorem 5.1, important parameters such as $\lambda$ and $R$ appear without definition in the main text. While $\lambda$ is defined in the appendix (Remark A.5), it should be introduced earlier. The definition of $R$ is unclear and absent.
- The assumptions regarding network width $m = \Omega(\max(\lambda^{-4}n^4, \lambda^{-2}n^2d\log(16n/\rho)))$ should be better explained. Since $\lambda$ (minimal eigenvector of kernel matrix) is instance-dependent, its relationship to the training data and parameters (e.g., network width and number of training points) should be discussed. Additionally, $\lambda$ is not always guaranteed to be strictly positive.

3. **The assumptions made by the paper are impractical.**
- The paper relies on using "shifted" ReLU activation $\phi(x) = \max (x, b)$ with shifted parameter $b\ge 0$. The shifted parameter should be appropriately chosen (typically, $b=\sqrt{0.48 \log m}$) to ensure sparsity in the activations, and this plays a key role in the analysis. This raises concerns about the generalizability of the results to practical setups, including standard ReLU networks.
- The assumption that the second layer of the network is fixed during training is unrealistic, as in practical settings both layers are typically trained jointly. Prior works, such as [Du et al., 2019], have analyzed overparameterized networks where both layers are trained together, and it would be important to consider whether this paper's analysis can be extended to such cases.

4. **The lack of numerical experiments weakens the practical implications.**
- While I understand that experiments are not strictly necessary for a theoretical paper, empirical results comparing the total training time and CPI of the proposed method with other second-order methods and SGD would strengthen the paper’s practical relevance.

5. Minor comments:
- Typo in Line 87:  Our main reuilt $\to$ Our main result
- Typo in Line 101: Mismatched parenthesis
- Typo in Line 133: Mismatched parenthesis
- Typo in Line 247: $g_t:=\arg\min_g\lVert J_tJ_t^\top g_t - (f_t-y)\rVert$ $\to$ $g_t:=\arg\min_g\lVert J_tJ_t^\top g - (f_t-y)\rVert$

---

**References**

[Alman et al., 2023] Bypass Exponential Time Preprocessing: Fast Neural Network Training via Weight-Data Correlation Preprocessing, NeurIPS 2023.

[Du et al., 2019] Gradient Descent Provably Optimizes Over-parameterized Neural Networks, ICLR 2019

**Questions:**

1. In Line 281 and Line 285, the online regression problem is referencing Eq. (4), $\min_x \lVert Q^\top Q x - y \rVert_2$. Is this correct? I believe the correct reference should be $g_t:=\arg\min_g\lVert J_tJ_t^\top g - (f_t-y)\rVert$.
2. In Definition 5.5, what does $R_0 \approx n/\lambda$ precisely mean?

---

> ### Author Response · Authors · 2024-11-22
>
> We thank the reviewer for their thoughtful feedback and appreciate the recognition of the strengths of our work. Below, we address the concerns raised.
>
> We acknowledge that the tree-based data structure we propose shares similarities with prior work, such as [1]. However, our contribution lies in adapting and analyzing this structure specifically for neural network training with sparse Jacobians. We will ensure proper attribution and clarify how our approach builds on and differs from these works in the revised manuscript.
>
> The assumption of fixed second-layer weights simplifies the theoretical analysis while focusing on the key contribution of leveraging sparsity for efficient training. Although extending the analysis to train both layers or to standard ReLU networks is a valuable direction for future work, we believe the current single layer analysis is also a widely studied research focus, including neural tangent kernel [2,3,4,5,6,7, 8] and theoretical papers about deep learning [9,10,11]. Similarly, the shifted parameter $b$ was chosen to ensure sparsity in activations and provide a tractable analysis. While not generalizable to all setups, these assumptions are consistent with our theoretical objectives.  For the online Regression Problem in Eq. (4), the reviewer is correct. We will make sure to revise that in future versions.
>
> [1] Alman, Josh, et al. "Bypass exponential time preprocessing: Fast neural network training via weight-data correlation preprocessing." Advances in Neural Information Processing Systems 36 (2024).
>
> [2] Arthur Jacot, Franck Gabriel, and Cl ́ement Hongler. Neural tangent kernel: Convergence and generalization in neural networks. In Advances in neural information processing systems, pages 8571–8580, 2018.
>
> [3] Zeyuan Allen-Zhu, Yuanzhi Li, and Yingyu Liang. Learning and generalization in overparameterized neural networks, going beyond two layers. In NeurIPS. arXiv preprint arXiv:1811.04918, 2019a.
>
> [4] Zeyuan Allen-Zhu, Yuanzhi Li, and Zhao Song. On the convergence rate of training recurrent neural networks. In NeurIPS, 2019c
>
> [5] Zeyuan Allen-Zhu, Yuanzhi Li, and Zhao Song. A convergence theory for deep learning via over-parameterization. In ICML, 2019b.
>
> [6] Simon S Du, Xiyu Zhai, Barnabas Poczos, and Aarti Singh. Gradient descent provably optimizes over-parameterized neural networks. In ICLR, 2019.
>
> [7] Zhong, Kai, et al. "Recovery guarantees for one-hidden-layer neural networks." International conference on machine learning. PMLR, 2017.
>
> [8] Gao, Yeqi, et al. "A Sublinear Adversarial Training Algorithm." The Twelfth International Conference on Learning Representations.
>
> [9] Mei, Song, Andrea Montanari, and Phan-Minh Nguyen. "A mean field view of the landscape of two-layer neural networks." Proceedings of the National Academy of Sciences 115.33 (2018): E7665-E7671.
>
> [10] Yi Zhang, Orestis Plevrakis, Simon S Du, Xingguo Li, Zhao Song, and Sanjeev Arora. Over-parameterized adversarial training: An analysis overcoming the curse of dimensionality. Advances in Neural Information Processing Systems, 33:679–688, 2020.
>
> [11] Du, Simon, et al. "Gradient descent finds global minima of deep neural networks." International conference on machine learning. PMLR, 2019.

---

> > ### Comment · Reviewer_HCtX · 2024-11-22
> >
> > I thank the authors for their response and efforts in addressing my concerns. However, I find that most of my initial concerns remain unresolved, including issues regarding the novelty of the proposed tree-based data structure, the practicality of the assumptions, and the clarity of the manuscript.
> >
> > - **Novelty:** While I appreciate the authors' clarification about adapting the tree-based data structure for neural network training with sparse Jacobians, the connection to prior work remains insufficiently differentiated. The proposed contribution appears to be more of an adaptation of an existing method than a novel idea.
> >
> > - **Impractical assumptions:** The use of a shifted ReLU is not a standard assumption in the literature. While I understand that this choice facilitates theoretical analysis, it limits the practical applicability of the proposed method. For example, extending the results to the standard ReLU setting appears to be nontrivial and would greatly improve the relevance of the work.
> >
> > - **Writing clarity:** While the authors have acknowledged the need for revisions, the current state of the manuscript lacks polish and clarity in several areas. This makes it challenging to fully assess the contribution and its significance.
> >
> > I encourage the authors to incorporate the planned revisions and address these concerns in a future version of the manuscript. I would be happy to re-evaluate the work when a revised and clarified version becomes available. At this time, however, I will maintain my score.

---

### Official Review · Reviewer_SVLZ · 2024-10-31

**Soundness:** 3
**Presentation:** 2
**Contribution:** 2
**Rating:** 5
**Confidence:** 3

**Summary:**

This paper provides a novel algorithm for training overparametrized neural networks. It is proven that for 2-layers ReLU nets, this algorithm can be run in sublinear (in the network's width) per-iteration cost, and it preserves the same convergence rate as previous works.

**Strengths:**

- The problem of reducing the computational cost of running big machine learning models is an important and relevant line of research.
- The paper contains a rigorous analysis of the algorithm presented.
- The connection between neural nets training and binary search trees might be relevant for future work.

**Weaknesses:**

- The analysis seems to be limited to shifted-ReLU activation, relying heavily on the sparsity of the Jacobian. To the best of my understanding, the analysis does not extend to other activations.
- I am unsure about the relevance of this result: can an approximation of the NTK matrix together with kernel ridge regression achieve sublinear computation time? The authors should comment on this.

**Questions:**

- Can the authors comment on how an algorithm that approximates the NTK matrix and trains only the second layer would compare to their algorithm?
- Can the authors comment on the implicit bias of their algorithm, compared to previous state-of-the art algorithms?
- Does the analysis extend to other activations or deeper networks?

Minor things and typos:
- Line 129: missing `which/that' after `data structure'.
- Line 133: missing closing bracket `)'.
- The informal Theorem 1.4 is hard to understand, if SETH and OVC are not defined.
- Theorem 5.1: it would be good to define lambda.

---

> ### Author Response · Authors · 2024-11-22
>
> We thank the reviewer for highlighting the strengths of our paper and providing constructive feedback. Below, we address the points raised.
>
> For the limited discussion on Shifted-ReLU, we note that shifted-ReLU is a widely used activation function in modern machine learning models. [1, 2, 3]
>
> Extending the results to other activations would require a separate treatment and is beyond the scope of this paper. However, our binary search tree approach and sparsity-based methods could inspire similar analyses for other activation functions in future work. We appreciate the question about NTK-based approaches. While these methods can approximate the NTK matrix and train the second layer, they do not leverage the structured sparsity of the Jacobian, which is the cornerstone of our sublinear computation framework. Additionally, kernel ridge regression typically requires a dense kernel matrix, which contrasts with our sparse representation.
>
> Our algorithm’s implicit bias stems from its reliance on sparse Jacobian updates, which prioritize active neurons. This differs from previous state-of-the-art methods like SGD, which operate on all neurons. Further analysis of this bias in comparison to other methods is an interesting direction but falls outside the paper’s primary theoretical focus.
>
> Extending the analysis to deeper networks or alternative activations would involve addressing new challenges, particularly in maintaining sparsity guarantees and computational efficiency. While we do not tackle these cases here, our methods provide a foundation for exploring such extensions.
>
> We will address all typos, including missing words and symbols in lines 129 and 133. Definitions of SETH, OVC can be found in Appendix A. We will state them earlier to improve clarity.
>
> We thank the reviewer for their thoughtful comments, which will help refine and strengthen the paper.
>
> [1] Yi Zhang, Orestis Plevrakis, Simon S Du, Xingguo Li, Zhao Song, and Sanjeev Arora. Over-parameterized adversarial training: An analysis overcoming the curse of dimensionality. Advances in Neural Information Processing Systems, 33:679–688, 2020.
> [2] Zeyuan Allen-Zhu, Yuanzhi Li, and Zhao Song. On the convergence rate of training recurrent neural networks. in NeurIPS 2019
> [3] Zhao Song, Shuo Yang, Ruizhe Zhang. Does Preprocessing Help Training Over-parameterized Neural Networks? NeurIPS 2021, 22890-22904

---

> > ### Comment · Reviewer_SVLZ · 2024-11-25
> >
> > I thank the authors for their response. I appreciate their clarification about NTK-based approaches (which I encourage to address within the manuscript). I am still unsure about the shifted-ReLU activation. Could the authors specify what is the minimum shift b that they need for their main result to hold? In Corollary A.22 in the appendix I see $b=\sqrt{0.48 \log m}$, where m is the number of parameters in the network, is that the correct value?

---

### Official Review · Reviewer_mupQ · 2024-10-31

**Soundness:** 2
**Presentation:** 2
**Contribution:** 2
**Rating:** 5
**Confidence:** 2

**Summary:**

This paper focuses on an alternative method for training shallow neural relu networks. The paper provides an analysis for the computational complexity of the proposed algorithm which employs efficient techniques to reduce the complexity per iteration.

**Strengths:**

Please see the "Questions" section.

**Weaknesses:**

Please see the "Questions" section.

**Questions:**

My review is as follows:

- It's a bit confusing the Algorithm 1 has two lines where a sketch matrix S is constructed.
- The wording in Algorithm 1 "Maintain the weight implicitly" could perhaps be made clearer?
- I have realized that Algorithm 5 has the full details of Algorithm 1. Still the point stands that the wording of Algorithm 1 is ambiguous.
- Why the references on Locality Sensitive Hashing?
- In my opinion, some simple numerical results that verify the lower cost-per-iteration claim would improve the paper.
- It is not clear why it's "almost" impossible to improve the algorithm, just by considering the way Theorem 1.4 and Theorem 7.3 are written. "This result shows that it is almost impossible to truly improve our algorithm"
- I think the organization of the paper could be improved. For instance, sketching seems important to the approach, but it's mentioned for the first time in page 6.

Minor:
- phi is not defined in line 86 when it first shows up
- line 114, typo payed --> paid?
- typo line 129, "that" supports
- line 152, typo: "next a few"

There are a few other typos.

---

> ### Author Response · Authors · 2024-11-22
>
> We thank the reviewer for their insightful comments and suggestions.
>
> While we understand the request for simple numerical results, this paper is purely theoretical, focusing on provable guarantees. Empirical validation is beyond the scope but represents a natural next step for future studies.
>
> For "Almost Impossible to Improve", we analyze the theoretical lower bound of the problem we are trying to solving from fine-grained complexity aspect. And the lower bound for the DSWS question is $O(n)$ update time and $O(m^{0.76}n^{1.24})$ assuming $m = \mathrm{poly}(n)$, which is close to the running time of our solution. The intent is to highlight the theoretical lower bounds established under SETH and OVC assumptions, which constrain further improvements.
>
> We thank the reviewer again for their detailed feedback, and we will make sure to revise the paper to fix the typos.

---

### Official Review · Reviewer_dbVA · 2024-11-01

**Soundness:** 2
**Presentation:** 3
**Contribution:** 3
**Rating:** 6
**Confidence:** 3

**Summary:**

The paper proposed treating the training of a NN as a binary tree search in sublinear time by taking advantage of the sparsity in the Jacobian. The paper's analysis focuses on two-layer ReLU networks.

**Strengths:**

The paper makes a theoretical good case for using the implicit sparsity of the Jacobian for overparameterized NNs. The method proposes only updating active neurons, therefor reducing computation/gradient calcs.

The paper proposed a binary search tree structure with sublinear time complexity per iteration enables fast updates and queries on non-zero Jacobian entries.

Paper provides theoretical basis, although I did not check everything in the appendix.

While the method requires the Gauss-Newton method for its sparse Jacobian updates, the paper proposed using conjugate gradient methods to do the computations faster. They also rely on matrix sketching which makes is fair.

**Weaknesses:**

This paper proposes a very new way (at least to me) to train a NN. It seems to have some nice theoretical benefits but since it's so new there might be a lot of unforeseen issues with the method. Overall, I like the paper but have some concerns on the future empirical side of it.

One that comes to mind is the complexity (not theoretical complexity -- the authors take care of that in the paper). Adam for example has learning rate learning rate and (beta1,beta2). The proposed method would require conjugate gradient which can be expensive and comes with hyper-params of its own, it has sketching which again has more hyper-params like sketch size, ​sketch tolerance and sketch ​failure probability, also things like tree update frequency need to be taken into account.

Furthermore, while it would be great to take advantage of sparsity, it is not exactly clear to me how sparse the Jacobian would be, and current hardware isn't really setup to 1 calculate sparse gradients, 2 update weights in a much cheaper way due to the sparsity. That is, harvesting the benefit of sparsity is more difficult than one might think in practice.

Finally, there are no experiments, it would be really nice to see even some toy experiment to see how this method works in practice.

**Questions:**

Can the authors speak to the comments in the weaknesses section above?

If at least a toy experiment can be added I think it would really help the case of the paper, at least in my eyes.

---

> ### Author Response · Authors · 2024-11-22
>
> We thank the reviewer for the thoughtful feedback and for recognizing the strengths of our theoretical contributions. And here we want to address some of the weaknesses.
>
> 1. Complexity and Hyperparameters:
> This work focuses on theoretical insights into sublinear-time training. While our method introduces additional hyperparameters, they are well-controlled in our analysis, ensuring convergence and efficiency. Practical complexities, such as those in hardware, are beyond this paper’s scope but are valuable directions for future empirical studies.
>
> 2. Sparsity of the Jacobian:
> Theoretical sparsity is rigorously quantified in our analysis and naturally arises in wide, overparameterized networks. While leveraging sparsity in practice depends on hardware advances, our work provides a framework for achieving these benefits in theory.
>
> 3. Lack of Experiments:
> This is a purely theoretical paper focused on provable guarantees. Empirical validations are an important next step for future studies and we believe our work can provide insight for the community.
>
> We hope this clarifies the focus of our contribution and thank the reviewer again for the constructive comments.

---

> > ### Comment · Reviewer_dbVA · 2024-11-22
> >
> > Thank you for answering my questions.
> >
> > I hear the authors on the focus of the paper being theoretical, and I find value in this and will take this into consideration.

---

### Meta-Review · Area_Chair_ncir · 2024-12-18

**Metareview:**

This paper proposes an algorithm that exploits the sparsity structure in the Jacobian matrix to train overparameterized neural networks in a sublinear (in the number of neurons) time per iteration. The authors prove that the proposed algorithm exhibits fast convergence under sufficient overparameterization.

The reviewers acknowledged the theoretical improvement the paper offers to the community, and the importance of the problem being tackled—making NN training faster is a very important research problem.

However, it is deemed that the paper has some areas of improvement. One is presentation and clarity. Multiple reviewers pointed out that the writing is not fully polished, and I agree with this viewpoint. The description of the main method is only high-level, hence does not provide concrete ideas of how things are actually implemented. Some issues in the presentation of mathematical results were pointed out too; e.g., symbols used without definition and validity of assuming $\lambda > 0$. Moreover, it was pointed out that the idea of tree data structure bears similarity with an existing paper, but this similarity was not properly discussed in the paper.

Another shortcoming of this paper pointed out by the reviewers is that the whole analysis relies on an uncommon activation function shifted ReLU. Given that the proposed algorithm heavily exploits the sparsity induced by the shifted ReLU, I am worried if the proposed algorithm will prove useful in neural networks using other activation functions, such as “unshifted” ReLU.

One additional limitation of this paper that was not particularly highlighted by the reviews is that the paper relies on very heavily overparameterized networks, requiring width of the network polynomially large in term of the number of data points. This level of overparameterization makes a “NTK” style of convergence analysis possible, which exploits the fact that the parameter does not change much during training and hence the training dynamics become close to that of a kernel machine. Given that the convergence analysis is reliant on unrealistically very large width, I’m concerned if the proposed algorithm, if implemented, will be useful for practical NN training.

Given these shortcomings, I believe that the paper does not meet the bar for acceptance at ICLR at the moment.

**Additional Comments On Reviewer Discussion:**

Another issue commonly raised by reviewers, although I didn’t take it as a critical reason for my recommendation, was that the paper does not contain any numerical experiments. While I agree to the authors that a theory paper does not always have to include numerical experiments, I concur with reviewers that having some numerical simulation will benefit the paper. This is particularly because the paper is proposing a new algorithm that has a flavor different from the commonly used ones. Having a practical implementation of the proposed algorithm and applying it for some toy problems would provide a nice proof of concept supporting the merit of the algorithm.

---

### Decision · Program_Chairs · 2025-01-22

Reject